# Individual and neighborhood based socioeconomic factors relevant for contact behaviour and epidemic control

Laura Di Domenico [1,3] ✉, Martina L. Reichmuth[1] & Christian L. Althaus[1,2]

## Abstract

**Background** Identifying sources of heterogeneity in contact patterns is key to inform disease transmission models. Recent works have investigated how individual-based socio-economic factors, besides age, affect contact behavior, but neglected the individuals' area of living. Here, we aim at estimating contact matrices stratified by both individual-based and area-based socio-economic factors.

**Methods** We use social contact data from Switzerland collected in 2021, combined with a neighborhood-based index of socio-economic position (SEP). Despite lacking socio-economic information on the contacts, we develop a method to reconstruct contact matrices fully stratified by age, education level, and SEP, with varying assortativity levels.

**Results** We find a positive association between education level and number of contacts in the elderly, and, notably, a negative association between SEP level and number of contacts in adults. Compared to homogeneous mixing, accounting for heterogeneous contact patterns leads to higher attack rates in groups with high education level, especially for adults living in low SEP areas and seniors living in high SEP areas. Adults and young individuals living in high SEP areas are the main contributors to transmission. Including socio-economic factors into model parameterization has limited effect on the basic reproduction number but substantially influences the effectiveness of control strategies. The more assortative contacts are, the higher the control effort required by a targeted strategy to be successful in preventing disease spread.

**Conclusions** Our results shed light on contact behavior in previously neglected socio-economic groups, enable model integration of socio-economic indicators, and provide insights to improve disease control.

## Plain language summary

Patterns of human interaction (also known as contact patterns) that impact disease spread can vary depending on age and education level, but less is known about the influence of the economic conditions of the area where people live. In addition, socio-economic information of the contacts (such as income, education or occupation) is often not collected in traditional surveys. In this study, we characterize mixing patterns across different socio-economic groups, even when some information is missing. Using mathematical modeling, we identify which groups generate most infections, and which groups experience the highest disease burden. We also show how diverse contact patterns can determine the success of a control strategy. Our findings can help improve the design of public health measures to contain epidemic spreading.

The COVID-19 pandemic has highlighted the value of infectious disease modeling incorporating data on human behavior, such as social contact patterns, to evaluate the effectiveness of control strategies and inform public health policies[1]. Several modeling works integrated age-stratified contact matrices[2–6], estimated from pandemic survey data[7,8]. However, contact patterns can vary not only by age but also by other variables, e.g., socio-economic factors (income, education level[9–11]), vaccination status[12–15], and vaccine hesitancy[16], that can all affect individuals' risk perception and adaptive behavior.

Population response to COVID-19 measures varied across socio-economic groups. Job market structure, income and house crowding acted as constraints in reducing mobility, shaping adherence to public health measures[17–19]. Socio-economic factors also influenced testing rates and response to change in testing guidelines[20,21]. Moreover, studies found socio-economic factors associated with COVID-19 disease severity[20,22] and with vaccine uptake, both for COVID-19[23–26] and other diseases[27,28], showing that individuals with lower socio-economic status are less likely to be vaccinated. These inequities call for the integration of socio-economic heterogeneity in epidemic models[29–33].

Recent epidemic modeling works included mixing patterns stratified by age and socio-economic variables[9,34,35]. However, these studies have focused on individual-based features of the survey participant (i.e., socio-

[1]Institute of Social and Preventive Medicine, University of Bern, Bern, Switzerland. [2]Multidisciplinary Center for Infectious Diseases, University of Bern, Bern, Switzerland. [3]Present address: Data Science Institute, Hasselt University, Hasselt, Belgium. ✉e-mail: laura.didomenico@uhasselt.be

economic status, or SES), neglecting the environment in which the individual lives (i.e., socio-economic position, or SEP). In Switzerland, a SEP index has been defined at building level[36,37], as an area-based measure reflecting the socio-economic situation of the neighborhood, intended as a collection of the closest 50 households. Previous work uncovered the association of the SEP index with health-related outcomes such as human papillomavirus (HPV) vaccine uptake[38], rate of testing, infection, hospitalization, and mortality related to COVID-19[20]. As the SEP refers to the neighborhood, rather than to individual-based characteristics (e.g., education level and income), it represents an additional dimension of socio-economic heterogeneity. By combining individual-based and area-based factors, it becomes possible to investigate contact behavior in intersectional socio-economic groups so far overlooked, e.g., individuals with low SES living in high SEP areas and vice versa.

To integrate socio-economic factors into epidemic models, one convenient framework is to introduce an age-stratified contact matrix that can be further stratified by additional dimensions, such as SES and SEP[34]. However, estimating such a matrix from survey data requires knowledge of socio-economic information related to both the participants and their contacts. The latter information is often missing in traditional social contact surveys, where participants only report the age of their contacts. This hampers the construction of a fully stratified contact matrix, and also opens questions about whether contacts along social dimensions are assortative, i.e., to what extent individuals engage contacts with individuals belonging to the same socio-economic group. Quantifying the assortativity of contacts is key to understanding the impact of socio-economic heterogeneity on contacts, and therefore on disease spread and epidemic control.

In this study, we aimed to extend traditional age-dependent social contact matrices with two complementary (individual-based and area-based) socio-economic dimensions. We used social contact data collected in 2021 in Switzerland with SES information for the survey participants, combined with a neighborhood-based SEP index. First, we characterized the dependence of contact activity on socio-economic factors, distinguishing between SES (focusing on education level) and SEP, to identify determinants influencing the number of contacts. Secondly, we built contact matrices accounting for age and socio-economic factors, by stratifying for both the participant and the contacts. To deal with missing socio-economic information on the peers, we developed a method to reconstruct fully stratified contact matrices based on the observed partial data, by preserving conditions on reciprocity and aggregation. This method allowed us to identify ranges of contact assortativity across socio-economic groups. Third, we integrated the generated contact matrices into a Susceptible-Infectious-Recovered (SIR) epidemic model, to identify the most relevant socio-economic groups in terms of disease burden and transmission. Finally, we tested the effectiveness of a targeted strategy aimed at reducing contacts or increasing vaccination uptake in a specific socio-economic group. Through the concept of the type-reproduction number, we identified which targeted control strategies would be effective, and we exposed the relation between contact assortativity and control effort.

We find that the number of contacts is associated with education level and SEP, but the direction and strength of the association vary across age groups. We show that the more assortative contacts are in the additional socio-economic dimensions, the lower the chances that a targeted control strategy would be effective in preventing epidemic spread, and the higher the control effort needed.

## Methods
### Social contact data
We used Swiss contact data collected during the COVID-19 pandemic through the European CoMix survey[7], containing individual-based socio-economic information (education level, income) and spatial information (municipality) for survey participants. The data are described in detail in a previous study[39] and a subset of variables are available on Zenodo[40]. To increase the sample size, we aggregated data from three contact surveys involving adults, carried out in January, June and December 2021, for a total

of 3552 adult participants. We also considered three contact surveys for individuals younger than 18 years, filled out by their parents, for a total of 910 minors.

All participants in the survey reported the following information: age, gender, region (urban or rural), country of birth, education level, household income, employment status, household size, COVID-19 vaccination status, municipality of residence. We divided the participants into four age groups, i.e., 0–14, 15–24, 25–64, and 65+ year olds. We refer to these age groups as children, young adults, adults, and seniors, respectively. We aggregated the education level into two categories: high education level, and middle-low education level. The former group corresponds to the tertiary level in the Swiss Education System[41], which includes advanced vocational education or a university degree (Bachelor, Master, or PhD); the latter group includes individuals with upper-secondary education or without any post-compulsory education. All children were classified as middle-low education. Further details about participant covariates and survey periods are contained in the Supplementary Information (Fig. S1). In a sensitivity analysis, we classified children based on the education level of the parents, to test the influence of their household on contact behaviour (Supplementary Information, Section 6).

Participants reported their contacts. A contact was defined as anyone who met the participant in person with whom at least a few words were exchanged or physical contact was made. The age of the contacts was available for 28,436 out of 35,072 contacts declared (81%). If a participant did not report information on the age of the contact, we imputed one of the four age groups (0–14, 15–24, 25–64, 65+ year olds), proportionally to the observed distribution of contacts' age for participants of the same group. Some participants reported contacts with age brackets overlapping two of the age groups under consideration (e.g., contacts declared as 0–18 year olds). As above, we sampled from the reported age group with a weighting consistent with the age distribution of contacts for the participants' own age group.

### SEP index
We used a neighborhood-based index of socio-economic position, so-called SEP, that has been defined for each residential building in Switzerland. The SEP index is a composite measure of the socio-economic level of the neighborhood (defined as the 50 closest households) based on four domains, i.e., rent per square meter, education level, occupation and overcrowding. For details on the construction of the index, see refs. 36,37.

We assigned a SEP level (low or high) to each participant based on the municipality of residence declared in the survey, with the following approach. First, we computed a weighted average SEP index at municipality level. We created an aggregated index weighted by the number of individuals living in each residential building in the municipality of interest, using household population data[42] (see Supplementary Information, Fig. S2 for details). Then, we used the weighted average SEP by municipality to group participants as low SEP or high SEP based on their municipality of residence, using the median as threshold value.

### Population data
We aimed at stratifying the Swiss population by age group (0–14, 15–24, 25–64, and 65+ year olds), education level (high or middle-low), and SEP level (high or low). As more detailed data were not available, we extrapolated an estimate combining data at district level by age group[43] or by education level[44], with national data by education level and age group[44]. Details are contained in the Supplementary Information (Fig. S3).

### Contact activity by group
We first computed the crude mean number of contacts per participant by education level (high or middle-low) and SEP (high or low), for each age group, after truncation at 50 contacts to remove the impact of outliers. We displayed the results in a heatmap to uncover the groups with higher or lower number of contacts. We stratified the analysis by age group (Fig. 2). Heatmaps further stratified by survey wave can be found in the

Supplementary Information (Fig. S4). We built bootstrapped confidence intervals for the mean number of contacts out of 1000 random samples with replacement of the participants data.

## Statistical analyses

We used a generalized linear model, to estimate the effect of an individual-based socio-economic factor, i.e., education level, and area-based SEP level on the number of contacts. We assumed a Poisson distribution to account for count data. In the main analysis, the response variable for the regression model was the overall number of contacts per participant. In a sensitivity analysis, we analyzed contacts outside the household, and contacts at work (Fig. S5). We adjusted for age group, panel wave, gender, region, country of birth, household income, household size, employment status, vaccination status, and type of day (weekday or weekend). In a sensitivity analysis, we also adjusted for population density (Fig. S6) and stratified by survey wave (Fig. S7). We included an interaction term between age and education level, and between age and SEP level, to account for effects stratified by age group. We then computed the combined relative risk (RR) as $\exp(\beta + \beta')$, where $\beta$ is the estimated coefficient for education level only (or SEP level only), and $\beta'$ is the estimated coefficient for the corresponding interaction term with age (Fig. 2). In a sensitivity analysis, we also included an interaction term between age and household income (Fig. S8).

## Age-stratified contact matrix

We estimated an age-stratified contact matrix from survey data[45], adjusting for population demography and contact reciprocity. We did not adjust for weekend effect, as the proportion of participants surveyed during weekdays or weekends was representative of the weekly pattern. We derived a contact matrix $M$ whose elements $M_{ij}$ represent the average number of contacts that one individual in age group $i$ engages with individuals in age group $j$, with $i, j$ being either 0–14, 15–24, 25–64, and 65+ year olds. We corrected for reciprocity by defining the matrix $M_{i,j}^{rec} = (N_i M_{ij} + N_j M_{ji})/2N_i$, to ensure the symmetry $N_i M_{i,j}^{rec} = N_j M_{j,i}^{rec}$, where $N_i$ is the population size of age group $i$, and $N_i M_{i,j}^{rec}$ is the total number of contacts between age group $i$ and age group $j$.

## Expanded contact matrices

We aimed to estimate an expanded reciprocal contact matrix $\widehat{M}_{s,t}$, with $s, t$ being of the form $(i, v, d)$, stratified by age group $i$, SEP level $v$ (low or high) and education level $d$ (middle-low or high). As SES indicator, we focused on education level only rather than household income, as stratified population data (i.e., number of individuals $N_s$ for any group $s = (i, v, d)$ in the general Swiss population) were only available for $d$ being education level and not for household income.

In the social contact survey, information on education level and SEP was available only for participants, while it was missing for contacts, for whom only age was available. We first built an intermediate contact matrix $\overline{M}_{(i,v,d),j}$ disaggregated on the participant side. This is a rectangular matrix informed from available data. The element $\overline{M}_{(i,v,d),j}$ represents the average number of contacts of a participant in age group $i$, SEP level $v$ and education level $d$, engaged with individuals belonging to age group $j$. We derived an adjusted contact matrix $\overline{M}^{rec}$ correcting by reciprocity, to be consistent with the reciprocal age-stratified contact matrix $M^{rec}$ after aggregating over socio-economic groups. Details on this reciprocity correction are presented in the Supplementary Information (Section 3.1).

We further expanded the adjusted intermediate matrix $\overline{M}^{rec}$ to a fully stratified contact matrix $\widehat{M}_{s,t}$. To derive the elements of the matrix $\widehat{M}$, we set a system of linear equations, with constraints due to the properties of the structure of the social contact matrix. In particular, the matrix $\widehat{M}$ is required to fulfill conditions on (i) reciprocity, i.e., the total number of contacts $N_s \widehat{M}_{s,t}$ between individuals in group $s$ and group $t$ must be equal to $N_t \widehat{M}_{t,s}$; (ii) aggregation, i.e., the number of contacts summed over all socio-economic groups (excluding age) must be consistent with the elements of

the adjusted intermediate contact matrix $\overline{M}^{rec}$, so that $\sum_{u,c} \widehat{M}_{(i,v,d),(j,u,c)} = \overline{M}_{(i,v,d),j}^{rec}$; (iii) positivity, i.e., the number of contacts in each cell cannot be negative.

Practically, the global system can be solved through 10 independent linear systems corresponding to specific blocks of the contact matrix (see Fig. S9 for an illustration). In order to solve the system for each block, it is required to set the value of some parameters $q \in (0, 1)$, that define the distribution of contacts across socio-economic groups (the mathematical definition is reported in the Supplementary Information, Section 3.2). They can be interpreted as assortativity parameters along SEP and education level dimension. Assortativity refers to the preferential behavior of an individual in engaging more contacts with people with the same characteristics. Age-stratified contact matrices estimated from empirical data are generally assortative with age[45], meaning that they have a stronger diagonal component compared to homogeneous mixing, where the number of contacts is proportional to the size of the contact group and no preferential behaviour is in place.

We explored the parameter space through uniform random sampling in the range $(0, 1)$ for each parameter $q$, and identified combinations of parameter values solving the system and leading to a positive expanded matrix. A set of combinations are displayed in Fig. S10–12. We generated 10,000 expanded contact matrices $\widehat{M}_{s,t}$ with varying assortativity parameters, that are all compatible with our partial data $\overline{M}_{(i,v,d),j}$. We then investigated the implications of these matrices in terms of population mixing and disease spreading. Moreover, we validated our reconstruction algorithm using a dataset of social contacts stratified by age and one socio-economic indicator (Supplementary Information, Section 5).

## Assortativity index

To characterize the mixing along the SEP and education level dimensions, we computed two assortativity indexes based on the 2×2 contact matrices $\widehat{M}_{v,u}$ and $\widehat{M}_{d,c}$ stratified by only SEP or only education level, respectively. These matrices are obtained from the expanded contact matrix $\widehat{M}$ after aggregation. The aggregation is performed by summing along the rows and computing a weighted mean across the rows based on the group size. As assortativity index $\alpha$, we decided to use the definition introduced in ref. 46 based on the trace of the matrix of proportions of contacts (the definition is reported in the Supplementary Information, Section 3.3). The index $\alpha \in (0, 2)$ is defined so that: (i) $\alpha = 2$ indicates fully assortative (or "segregate") mixing, where individuals are in contact only with people in their same group, i.e., no contacts across different groups occur (corresponding to null matrix elements outside of the diagonal); (ii) $\alpha = 1$ indicates homogeneous or proportional mixing[47], i.e., the per-capita contact rate $\widehat{M}_{v,u}/N_u$ is constant for any group $u$, and therefore the number of contacts $\widehat{M}_{v,u}$ is proportional to the size of the contacted group $N_u$; (iii) $\alpha = 0$ represents fully disassortative mixing, i.e., no contacts within the same group occur (null matrix elements on the diagonal). We computed the assortativity index $\alpha$ for both $\widehat{M}_{v,u}$ and $\widehat{M}_{d,c}$ obtained from 10,000 different expanded contact matrices, and we compared it with an expanded contact matrix $\widehat{M}_{\mathrm{hom}}$ that assumes homogeneous mixing along SEP and education levels.

## Transmission model

We considered an extended version of an age-stratified Susceptible-Infectious-Recovered (SIR) compartmental model for disease transmission[48,49]. Assuming the same epidemiological parameters for all age groups, the force of infection on susceptible $S_i$ of age group $i$ can be written as $\lambda_i = \beta \sum_j M_{i,j} I_j/N_j$, where $\beta$ is the transmission rate per contact, $M_{i,j}$ is the age-stratified contact matrix introduced above, and $I_j/N_j$ is the fraction of infectious individuals in age group $j$. Here, we extended the age stratification to a stratification by SEP and education level, where $S_{i,v,d}$ is the number of susceptible of age group $i$, SEP level $v$, education level $d$. Analogously, we integrated our estimated expanded contact matrix $\widehat{M}_{(i,v,d),(j,u,c)}$ in the force of infection $\lambda_{i,v,d} = \beta \sum_{j,u,c} \widehat{M}_{(i,v,d),(j,u,c)} I_{j,u,c}/N_{j,u,c}$.

## Reproductive number and disease spread

The basic reproductive number $R_0$, defined as the average number of infections generated by a case in a fully susceptible population, is a key quantity which characterizes the epidemic dynamics[50]. For structured compartmental models, $R_0$ can be computed as the dominant eigenvalue $\rho(K)$ of the so-called next-generation matrix $K$[51,52], which encodes the contact matrix $M$ and epidemiological characteristics of the groups. When all groups are epidemiologically equivalent (i.e., having the same susceptibility, infectiousness, and recovery rate), the ratio $\rho(K')/\rho(K)$ is equal to $\rho(M')/\rho(M)$. Hence, given a couple of contact matrices $M$ and $M'$, the relative variation in their dominant eigenvalue would correspond to the relative variation in $R_0$ when integrating these matrices into the *SIR* model. We computed the relative variation in $R_0$ for a set of expanded contact matrices $\widehat{M}$, compared to a matrix $\widehat{M}_{\mathrm{hom}}$ with homogeneous mixing in the SEP and education level dimension (i.e., number of contacts is proportional to the size of the contact group). We also computed the relative variation in $R_0$ when adding one dimension to age-stratification, either SEP or education level only, to quantify their relative role. We expect this relative variation to be always positive (see ref. 34 and Supplementary information, Section 3.4). For each group $s = (i, v, d)$, we also computed its proportional contribution to $R_0$. In particular, for each group $s$ we computed the cumulative elasticity $e_s$, following the framework described in ref. 53. The elasticity index allowed us to identify which groups contribute the most to transmission. Finally, we simulated the epidemic spread in absence of control measures and computed the relative attack rate in each group (i.e., the cumulative number of infections at the end of the epidemic), to identify the ones mostly affected by disease burden. We considered the same epidemiological parameters for all socio-economic groups, to isolate the effect of contact patterns on disease spread.

## Type-reproduction number and targeted control strategies

In a population with homogenous mixing, the basic reproductive number is linked with the herd immunity threshold $1 - 1/R_0$, which represents the fraction of susceptible required to be fully immune to halt the epidemic[54]. For heterogeneous mixing, the type-reproduction number $T_g$ has been introduced as a measure of the control effort needed to contain the epidemic when targeting one specific group $g$ of the population[55,56]. The epidemic is controlled if a proportion of group $g$ greater than $1 - 1/T_g$ is permanently immune or fully isolated at the start of the epidemic. Thus, the quantity $1 - 1/T_g$ corresponds to the immunity threshold specific to some target group(s). The type-reproduction number can be computed from the next-generation matrix (for additional details, see refs. 55,56 and Supplementary Information, Section 4.1). We computed the type-reproduction number and the corresponding immunity threshold for a set of target groups, to test

the effectiveness of targeted strategies. For the strategies allowing effective control of the epidemic, we computed the overall control effort $(1 - 1/T_g) * N_g/N$, i.e., the fraction of targeted individuals over the whole population, to allow comparisons across strategies targeted to groups of different sizes.

## Statistics and reproducibility

Analyses were conducted using R (version 4.3.2) and Python (version 3.8.5). We generated a random sample of 10,000 synthetic expanded contact matrices. All results are reproducible using the code provided on GitHub here[57].

## Ethics statement

The CoMix study protocols and questionnaires were approved by the local ethics committee of the Canton of Bern (Gesundheits-, Sozial- und Integrationsdirektion, Kantonale Ethikkommission, project number 2020–02926). Informed consent was obtained from all adult participants, and from parents or legal guardians in the case of minors. All methods were performed in accordance with regulations.

# Results

## Contact activity by socio-economic group

For our analysis, we divided each age group in four socio-economic groups, i.e., individuals with high or middle-low education living in areas with high or low SEP. The size of each group in the general Swiss population is displayed in Fig. 1a. Higher densely populated areas typically have a higher average SEP compared to lower densely populated areas (Fig. 1b). In absence of information on the residential address, participants in the survey were classified as having low or high SEP based on the average SEP of the declared municipality of residence (see Methods). Participant sample sizes were found to be well representative of the general population (Fig. S2).

We computed the mean number of contacts per participant for each socio-economic group, distinguishing by age (Fig. 2a–d). Young adults (15–24 year olds) living in low SEP areas had more contacts compared to those with high SEP. Among adults (25–64 year olds), the group with the highest contact activity were individuals with high education level living in low SEP areas. This was not observed for seniors (65+ year olds), where individuals with high education level engaged a similar number of contacts regardless of their SEP. The different contact patterns observed within each age group suggested a possible interaction effect between age and socio-economic factors in shaping contact activity.

We included an interaction term with age for SEP and education level in a multivariate regression model, adjusting for household income, household size, employment status and other relevant variables (see

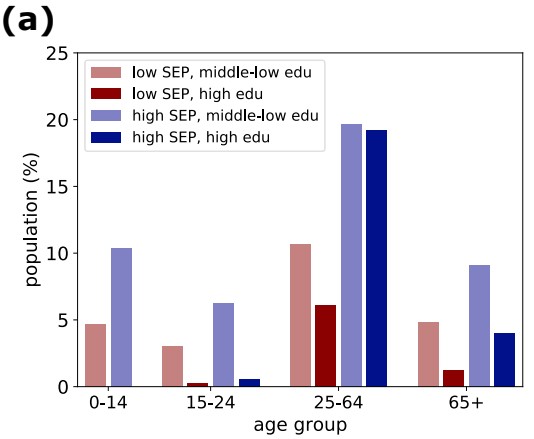

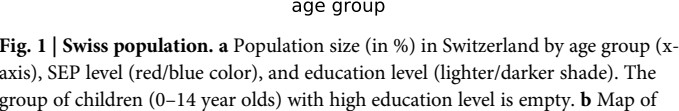

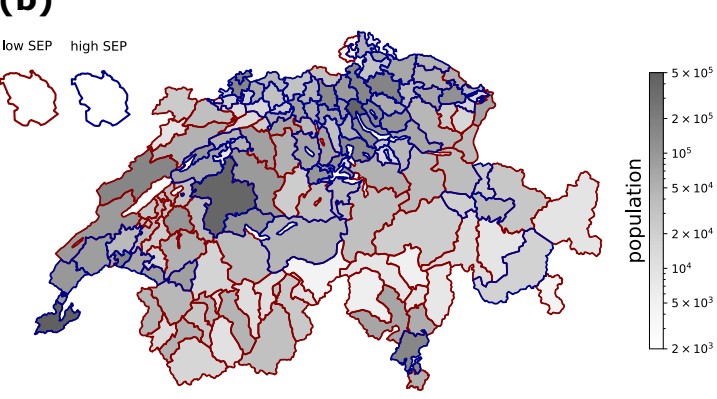

**Fig. 1 | Swiss population. a** Population size (in %) in Switzerland by age group (x-axis), SEP level (red/blue color), and education level (lighter/darker shade). The group of children (0–14 year olds) with high education level is empty. **b** Map of

population size in Switzerland at district level. The face color intensity of each district indicates the population. The edge color of each district indicates the SEP classification (either high SEP and low SEP, in blue and red, respectively).

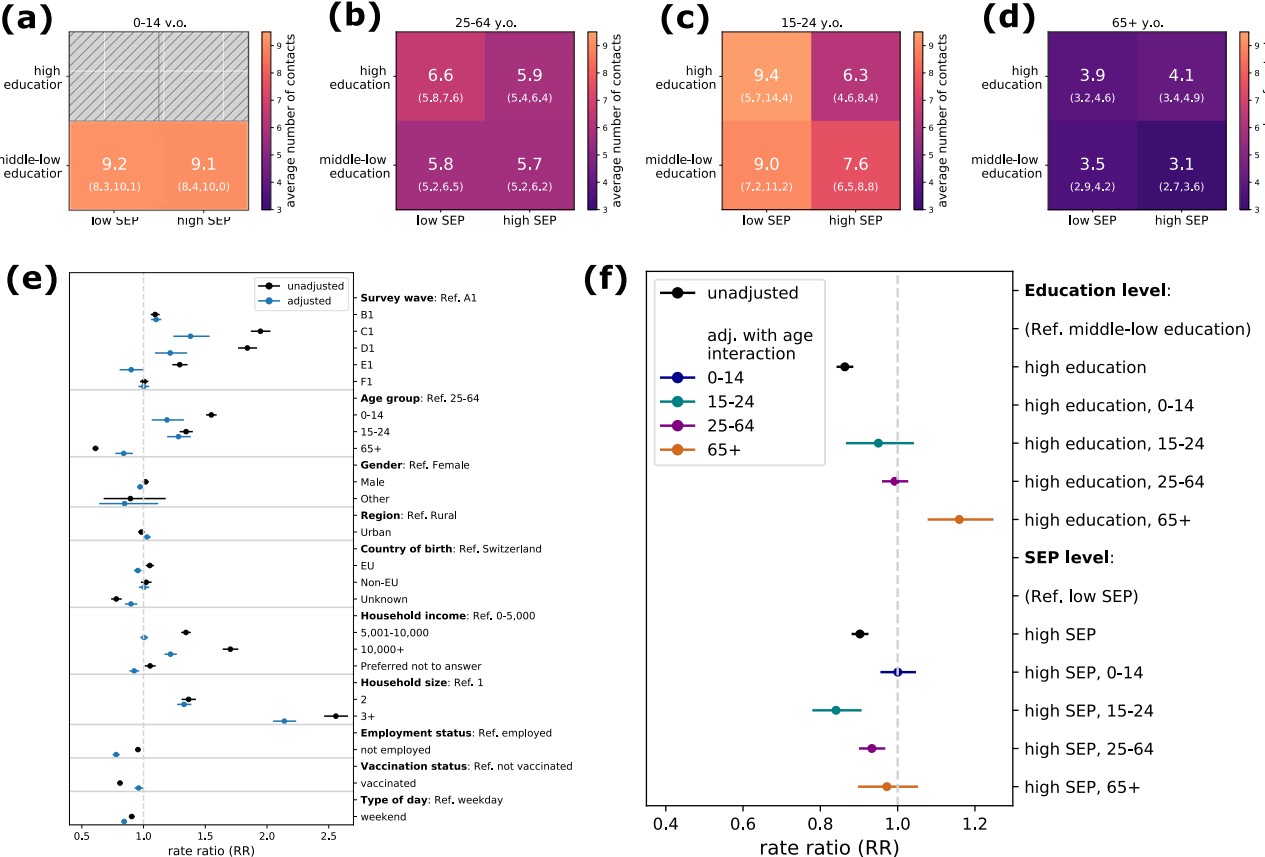

**Fig. 2 | Determinants of contact behaviour. a–d** Average number of contacts for each individual-based (education level) and area-based (SEP) socio-economic group. The heatmap shows the crude mean number of contacts (along with 95% confidence interval obtained with 1000 bootstraps) engaged by a participant belonging to one of the four socio-economic groups, depending on the age group (**a**) 0–14 year olds (children), (**b**) 15–24 year olds (young adults) (**c**) 25–64 year olds (adults) and (**d**) 65+ year olds (seniors). **e, f** Results of the regression analysis. The

rate ratio (RR) represents the relative change in the outcome, i.e., the average number of contacts of a participant in a given group compared to a group of reference. Black and colored dots represent the estimate obtained with a univariate and multivariate model, respectively. Bars indicate 95% confidence intervals; sample size $n = 3552$ participants above 18 y.o., and $n = 910$ participants below 18 y.o. The notation of the survey waves (top of **e**) is explained in Fig. S1.

Methods and Fig. 2e). On one hand, we found that the number of contacts was positively associated with individual-based education level in seniors (rate ratio RR = 1.16, 95% CI (1.08–1.25) for high education level compared to middle-low), while adults and young adults showed a similar number of contacts regardless of education level (RR = 0.99 (0.96–1.03) and RR = 0.95 (0.87–1.04), respectively, Fig. 2f). In a sensitivity analysis distinguishing by survey wave, we found that the association between number of contacts and education level was always positive for seniors, while for adults and younger individuals the sign of the association varied (Fig. S7). Moreover, household income was also found to be positively associated with the number of contacts (Fig. 2e), and the positive association was also found for all age groups after inclusion of an interaction term (Fig. S8).

On the other hand, for some age groups we found a negative association between the number of contacts and SEP level. Adults with high SEP had fewer contacts compared to low SEP (RR = 0.93 (0.90–0.97), Fig. 2f). This effect was even stronger in young adults (RR = 0.84 (0.78–0.91)), while no significant effect was found in children and seniors (Fig. 2f). Similar results were found when restricting to contacts outside home (Fig. S5), and when stratifying by survey wave (Fig. S7). Finally, we found that contacts significantly increased with household size, as expected (Fig. 2e). Fewer contacts were estimated during weekends compared to weekdays (RR = 0.84 (0.82–0.87)), and for unemployed individuals compared to employed (RR = 0.78 (0.75–0.81)), while the number of contacts did not vary significantly with gender, region, country of birth and vaccination status (Fig. 2e).

Overall, these results highlight heterogeneous contact activity across age and socio-economic groups, calling for the need to build a social contact matrix stratified by age, SEP, and SES. In the rest of the analysis, we focused on education level as SES indicator, as population data were not available for other SES indicators (see Methods).

### Expanded contact matrices

To build a fully stratified contact matrix, we developed a method to synthetically extend the available empirical data, as information on SEP and education level of contacts was missing. We first estimated an age-stratified contact matrix (Fig. 3a), adjusted for reciprocity, using a standard approach (see Methods). Then, we estimated an intermediate contact matrix stratified by SEP and education level on the participant side, using the information on education level and SEP provided in the survey (Fig. 3b). We applied a reciprocity correction to ensure consistency with the age-stratified contact matrix. The cumulative number of contacts for each group relative to the total number of contacts is shown in Fig. 3c.

Our method allows us to extend the intermediate contact matrix by distributing the age-specific number of contacts across the four missing socio-economic groups, while preserving conditions on aggregation and reciprocity of the expanded contact matrix (Methods). We used our method to generate 10,000 expanded contact matrices, which differ according to the choice of parameter values defining the level of assortativity across the socio-economic groups (see Methods and Supplementary Information for the definition and the interpretation of these parameters). One realization of a synthetic matrix generated through our method is displayed in Fig. 3d. We

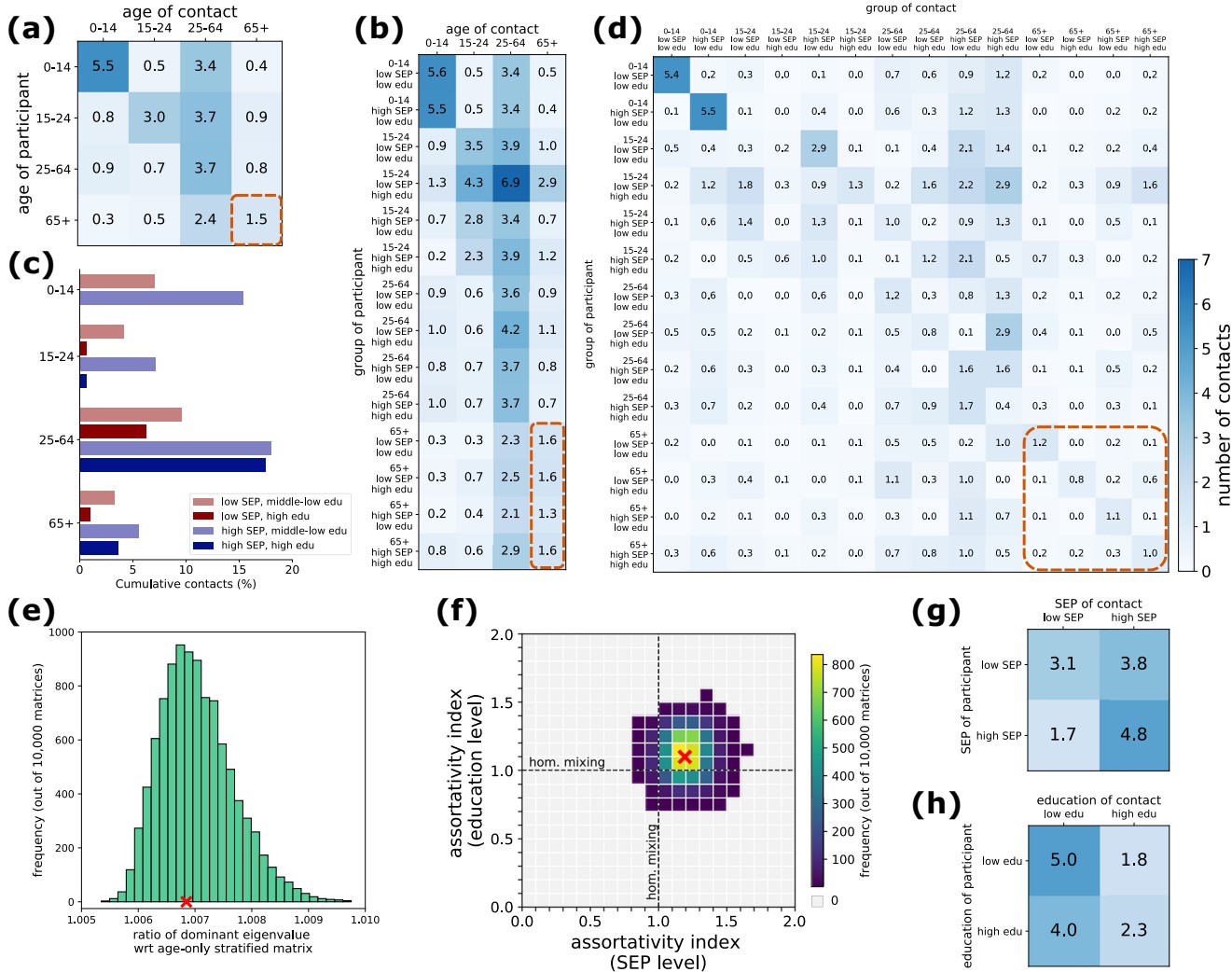

**Fig. 3 | Expanded contact matrices. a** Age-stratified contact matrix $M^{rec}_{i,j}$, adjusted for reciprocity, informed from the available data. **b** Intermediate contact matrix $\bar{M}^{rec}_{(i,v,d),j}$ stratified by age, SEP and education level on the participants' side, adjusted for reciprocity, informed from the available data. **c** Cumulative number of contacts (in %) engaged by a group, out of the total number of contacts, computed as $\left(\sum_j N_s \bar{M}^{rec}_{s,j}\right) / \left(\sum_{s,j} N_s \bar{M}^{rec}_{s,j}\right)$ for each group $s = (i,v,d)$ with age $i$, SEP level $v$, and education level $d$. **d** One example of a fully expanded synthetic contact matrix $\hat{M}_{(i,v,d),(j,u,c)}$, compatible with the age-stratified matrix $M^{rec}$ and with the inter-mediate matrix $\bar{M}^{rec}$ under aggregation. The matrix $\hat{M}$ was obtained with a given combination of the free assortativity parameters (parameter values are reported in the Supplementary Information, Table S1). The dashed orange rectangles across panels (a), (b) and (d) indicate an example of matrix elements that are related through aggregation and stratification (in this case, mixing within the group of

seniors 65+ year olds). **e** Distribution of the ratio of the dominant eigenvalue of the expanded contact matrices compared to a matrix with homogeneous mixing in the SEP and education level dimensions. **f** Heatmap of the frequency of values of the assortativity index in the two dimensions (education level on y axis and SEP level on x axis). The assortativity index is defined so that a value equal to 1 indicates homogeneous mixing (i.e., mixing proportional to group sizes), and values larger and lower than 1 indicate more assortative and less assortative mixing, respectively, compared to homogeneous mixing. Gray cells indicate ranges of assortativity not observed in our set of expanded contact matrices. In both (**e**) and (**f**), we show the results for 10,000 synthetic matrices. The values of assortativity index and dominant eigenvalue ratio for the example of expanded contact matrix shown in **c** are shown with a red cross in (**e**) and (**f**). **g** Matrix shown in (**d**) stratified by SEP level only. **h** Matrix shown in (**d**) stratified by education level only.

then characterized the properties of the generated contact matrices in terms of $R_0$ and assortativity index, compared to a matrix neglecting stratification by SEP and education level.

First, for each expanded contact matrix, we computed the ratio of the dominant eigenvalue compared to the age-only stratified contact matrix (equivalent to an expanded contact matrix assuming homogenous mixing in the SEP and education level dimensions). We found that the ratio is always positive, meaning that including additional dimensions would increase the basic reproductive number $R_0$ (as expected, see Methods and ref. 34). However, the estimated relative variation is relatively small, ranging between 0.5% and 1% (Fig. 3e). By aggregating over one of the two social dimensions, we found that, compared to age-stratified mixing, the social dimension contributing the most to the relative change in the dominant eigenvalue (and therefore in $R_0$, assuming all groups are epidemiologically

equivalent) is the education level (Fig. S14). Compared to a model with full homogeneous mixing, the relative variation in $R_0$ obtained after introducing the age dimension is much higher than the variation obtained by integrating SEP or education level or the two combined (Fig. S15), suggesting that age is still the main driver of contact heterogeneity.

Then, for each expanded contact matrix, we also computed two aggregated matrices, stratified only by SEP level (Fig. 3g) or by education level (Fig. 3h) and extracted an assortativity index, a summary measure of assortativity for each social dimension (Fig. 3f). The assortativity index ranges between 0 and 2, with a value of 1 indicating homogeneous mixing, where contacts between socio-economic groups occur at random in proportion to group sizes. Values above or below 1 indicate assortative mixing or disassortative mixing, respectively, depending on whether contacts occur more frequently within the same socio-economic group or between groups

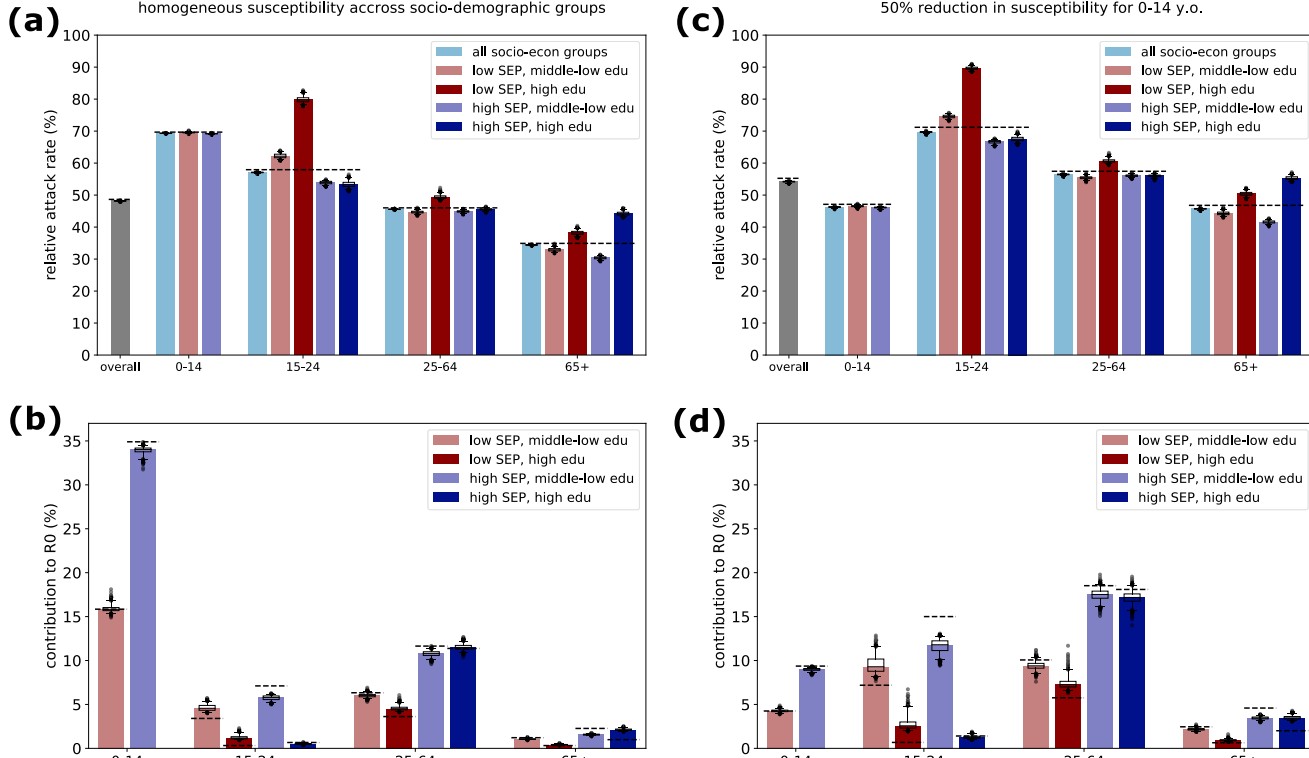

**Fig. 4 | Epidemic spread in absence of control strategies.** We considered a structured SIR epidemic model, stratified by age and socio-economic levels with heterogeneous mixing, and two epidemic scenarios: homogeneous susceptibility and infectiousness across age and socio-economic groups (panels a-b), or a reduction in susceptibility of 50% for children (panels c-d). In both scenarios, we assumed $R_0 = 1.5$, and an average infectious period of 3 days. **a** Relative attack rate for each group, i.e., fraction of the cumulative number of infected individuals over the size of the group. Color indicates the low SEP (red) and high SEP group (blue); color shade indicates groups with middle-low education level (lighter shade) and high education level, respectively (darker shade). The attack rate overall (gray) and aggregated by age groups (light blue) are shown for comparison. Horizontal dashed lines indicate results obtained from a reference epidemic model assuming an age-stratified matrix

with homogeneous mixing in SEP and education level. **b** Proportional contribution to $R_0$ of each group, computed using the framework of perturbation analysis of the next-generation matrix introduced in ref. 53. Horizontal dashed lines indicate results obtained from a reference epidemic model assuming an age-stratified matrix with homogeneous mixing in SEP and education level, where differences in contribution to $R_0$ between socio-economic groups in a given age class are only driven by differences in group sizes, and not by differences in contact activity. **c–d** As in (**a–b**), but assuming a lower susceptibility for the youngest age group of children (0–14 y.o) with a 50% reduction compared to the other groups. In all (**a–d**), the bars represent median values; the whiskers represent the 2.5 and 97.5 percentiles computed across $n = 1000$ realizations of the expanded synthetic contact matrix.

with different socio-economic levels. Values equal to 2 or 0 indicate fully assortative mixing (i.e., only within-group contacts, corresponding to a diagonal matrix) or fully disassortative mixing (i.e., no within-group contacts, corresponding to an empty diagonal), respectively. On one hand, we found that, in the education level dimension, contacts can be more assortative or less assortative compared to homogeneous mixing (81% and 19% of matrices with assortativity index above or below 1, respectively). On the other hand, in the SEP dimension, contacts are almost always more assortative than homogeneous mixing (96% of matrices with assortativity index above 1), meaning that people with a given SEP level tend to have more contacts with people with the same SEP level. In both dimensions (SEP and education level), we found that, on average, the higher the assortativity, the higher the relative change in the dominant eigenvalue (Fig. S13).

Overall, our method allows us to restrict the two-dimensional space of assortativity to values compatible with the observed empirical data. As a validation, we applied our method on a dataset of social contacts stratified by age and one socio-economic dimension, for which assortativity was known. We showed that our method successfully reconstructs a range of assortativity that includes the true value. Results are shown in Figs. S17–S19 in the Supplementary Information.

### Impact on epidemic spreading

We integrated our expanded contact matrices into an SIR epidemic model, to investigate how heterogeneous contact patterns along the education level

and SEP dimensions can impact disease spreading. We considered two scenarios, namely (i) a scenario where all groups are epidemiologically equivalent, and (ii) a scenario where children (<15 year olds) have a reduced susceptibility compared to other age groups (50% reduction), to mirror a COVID-19-like scenario[58]. We characterized disease spreading by identifying which groups generate most infections, and which groups experience the highest infection burden. First, to identify which groups are mostly affected by the disease, we computed the relative attack rate, i.e., the cumulative number of infections within each group. Second, to identify which groups contribute the most to transmission, we computed the relative contribution to $R_0$ of each group, using the framework introduced in ref. 53. This index takes into account the total contact activity of each group, the mixing with other groups, and the sizes of the groups, together with their epidemiological characteristics (see Methods for additional details).

We found that, despite the small relative variation in $R_0$ compared to homogeneous mixing (Fig. 3e), accounting for additional social dimensions in the contact matrix allows to uncover heterogeneity in disease burden and contribution to transmission across groups (Fig. 4). Among the adult population (25–64 year olds), the most affected group consisted of individuals with high education and low SEP, who faced more infections than the average adults (median relative attack rate 49% compared to average 46% in scenario (i), Fig. 4a; 61% compared to 56% in scenario (ii), Fig. 4c). This discrepancy is even stronger among young adults (15–24 year olds), with median relative attack rate 80% in the group with low SEP and high

education, compared to average 57% in young adults in the first scenario (90% compared to 70% in scenario (ii)). Instead, among seniors (65+ year olds), the most affected group corresponded to individuals with a high education level living in high SEP areas (median relative attack rate 44% compared to 34% in scenario (i); 55% compared to 46% in scenario (ii)). We found that the major contribution to $R_0$ was provided by individuals living in high SEP areas, in particular children and adults in the scenario of homogeneous susceptibility (contributing together to 56% of $R_0$, Fig. 4b), and adults and young adults in the scenario with reduced susceptibility for children (total contribution 48%, Fig. 4d).

### Effectiveness of targeted control strategies

Next, we evaluated the effectiveness of control strategies. Here, we focused on targeted control strategies, i.e., strategies aiming at one group, while the rest of the population is unaffected. In a simplified scenario, we assumed that, at the start of the epidemic, a fraction of one targeted group $g$ is permanently immune (e.g., through vaccination) or isolated due to a reduction in contacts. We determined whether there is a critical control effort (i.e., a critical fraction of the group computed as $1 - 1/T_g$ from the type-reproduction number $T_g$, see Methods) which would prevent the spread of the epidemic in the full population. If so, we then computed the corresponding overall control effort defined as $(1 - 1/T_g)*N_g/N$ where $N_g$ is the population size of the group and $N$ is the total population. The overall control effort thus represents the fraction of individuals targeted (belonging to the same group $g$) over the total population, and can be compared across strategies targeted at different groups. For each targeted strategy, we tested different expanded contact matrices with heterogeneous mixing, to investigate the interplay between assortativity and control effort. Given $R_0 = 1.5$, the estimated immunity threshold would be 33% under the assumption of a homogenous population (with full random mixing).

We considered here the epidemic scenario with reduced susceptibility for children, to model a disease similar to COVID-19[58]. Results for the homogeneous epidemic scenario are included in the Supplementary Information (Fig. S16). We tested three ways of partitioning the population. We partitioned the population in two groups, either by SEP level only (31% and 69% for low SEP and high SEP, respectively), or by education level only (69% and 31% for middle-low and high education, respectively), or in four groups based on both SEP and education level. Sizes of the groups are reported in Fig. 5j.

When considering groups based only on SEP level, we found that targeting individuals with high SEP (the largest group) is successful in 31% of the cases (Fig. 5i), depending on the expanded contact matrix, and the overall control effort required would range between 34% and 69% (Fig. 5a). The lower the assortativity index in the SEP dimension, the lower the control effort required (Fig. 5c). When considering individuals with low SEP as a target, we found very few contact matrices (1.2%) which would allow epidemic control. They correspond to very low assortativity levels in the SEP dimension (Fig. 5b), but if effective they would require a much lower control effort (27% to 31%, Fig. 5a).

Similar considerations hold when targeting individuals with high or middle-low education (Fig. 5d), noting that in this case the largest group is represented by individuals with middle-low education (which include all children aged 0–14). The control strategy would be more effective (i.e., it would require a lower control effort) when contact patterns are similar to or less assortative than homogeneous mixing in the education level dimension (Fig. 5e, f), analogously to the results found for the SEP dimension. The probability of success of a strategy targeted at individuals with high education is very low (less than 1% of the matrices), given the limited size of this group (31% of the total population, with no children included). In a sensitivity analysis where children were classified based on the education level of their parents rather than being all classified as middle-low education, strategies targeted at the high education level group would be more effective (Fig. S24).

Finally, we considered a partitioning in four groups as possible targets, i.e., individuals with low SEP and middle-low education, low SEP and high

education, high SEP and middle-low education, high SEP high education (Fig. 5g, h). The type-reproduction number indicated that in this case only strategies targeted at the group with high SEP and low education (the largest group, accounting for 45% of the population, Fig. 5j) can be effective in controlling the epidemic. A strategy targeted to this group would be effective for 29% of the matrices considered (Fig. 5i). In the subset of matrices for which epidemic control was possible, the overall control effort required ranges between 33% and 45% (Fig. 5g). We found that a lower control effort would be needed when contacts are less assortative both in the SEP and education level (Fig. 5h), combining the effects observed with the previous independent strategies considering only education level or only SEP stratification. Overall, we found that the more assortative contacts are in the additional socio-economic dimensions (SEP and education level), the lower the chances that a targeted control strategy would be effective in preventing epidemic spread, and the higher the control effort needed.

## Discussion

Social disparities observed in health-related outcomes highlight the need to integrate socio-economic heterogeneity in traditional age-stratified epidemic models[29–33]. In this study, we used survey data to characterize the number of contacts and mixing patterns across age and socio-economic groups, including both individual-based and neighborhood-based socio-economic factors. We found that the number of contacts varied with education level and SEP level, depending on the age group. Despite missing socio-economic information on peers, we used information on survey participants to reconstruct contact matrices fully stratified by age, SEP, and education level, by varying the distribution of contacts across different socio-economic groups, while ensuring conditions on aggregation and reciprocity of the mixing matrix. This method allowed us to restrict the boundaries of contact assortativity along social dimensions, identifying ranges compatible with the observed contact data. By integrating the extended contact matrices into a transmission model, we identified the socio-economic groups that contribute the most to transmission and that experience the heaviest disease burden. We showed that contact assortativity determines the effectiveness of targeted control strategies, as higher levels of assortativity require larger control efforts in specific subgroups to prevent an epidemic outbreak.

Regarding education level, and considering all contacts irrespective of the location, we found that seniors with higher education level reported a higher number of contacts compared to seniors with middle-low education level. In a sensitivity analysis, this trend was detected both in adults and seniors when restricting to contacts engaged in locations outside the household (Supplementary Information). Also, household income was positively associated with the number of contacts. We considered contact surveys collected in Switzerland in January, June, and December 2021. Our results are in line with studies from other countries showing that individuals with higher education level had a higher number of non-household contacts during the COVID-19 pandemic[9,10,59], except for periods with more stringent measures where the opposite trend was observed[10,11]. The latter suggests that these socio-economic groups were better able to adapt to the epidemiological situation, by reducing and resuming their contacts according to non-pharmaceutical interventions[9].

Besides individual-based socio-economic factors that are commonly taken into consideration when analysing social contacts, our work also considers a neighborhood-based socio-economic factor (socio-economic position, SEP). We found that adults and young adults living in areas with a higher SEP reported fewer contacts than those living in low-SEP areas. This trend was consistent across survey waves and also robust for non-household contacts. In contrast with the positive association found for education level and income, these results suggest that neighborhood-based socio-economic factors encode different information on contact behavior compared to individual-based factors. They also suggest that the relationship between socio-economic factors and number of contacts may depend on the age group, with individual-based factors (education level) being more relevant for seniors, and area-based factors (SEP) being more relevant for adults and

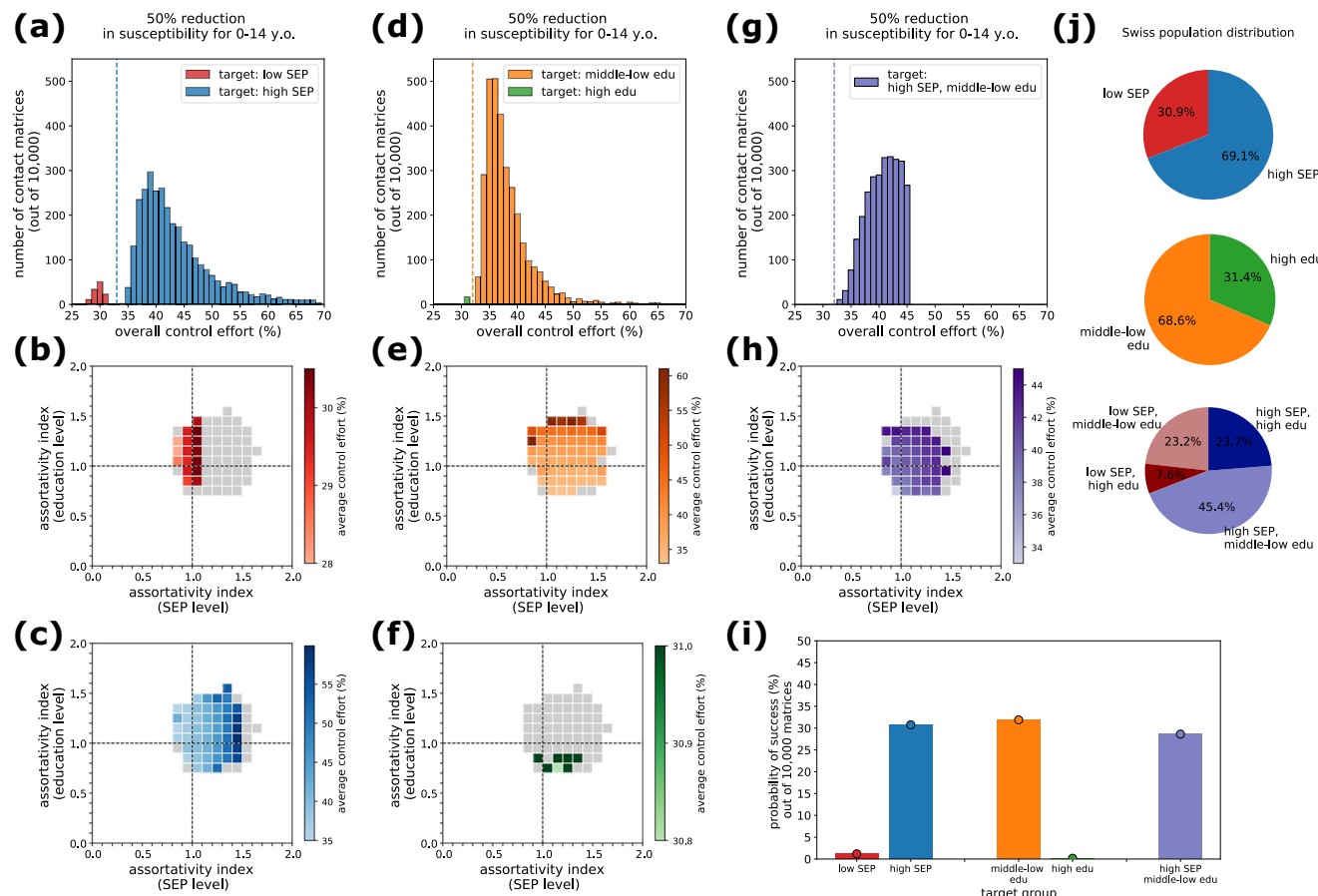

**Fig. 5 | Effective targeted control strategies.** Results for the epidemic scenario with reduced susceptibility for children. **a** Distribution of the overall control effort required by strategies targeted at individuals with low SEP (red) or with high SEP (blue), in the subset of matrices for which control was possible. Vertical dashed line represents control effort obtained from an epidemic model assuming an age-stratified matrix with homogeneous mixing in SEP and education level. **b** Assortativity levels in the SEP and education dimensions for the subset of matrices which allow effective control (colored cells) and those matrices for which the strategy would not be effective (gray cells). The color gradient indicates the average control effort required for a given range of assortativity. The strategy considered here is targeted at the low SEP group. **c** As in panel (**b**), but considering a strategy targeted at the group with high SEP. **d** Distribution of the overall control effort required by strategies targeted at individuals with middle-low (orange) or with high education level (green). **e** As in panel (**b**), but considering a strategy targeted at the group with middle-low education. **f** As in panel (**b**), but considering a strategy targeted at the group with high education. **g** Distribution of the overall control effort required by strategies targeted at individuals with high SEP and middle-low education level. **h** As in panel (**b**), but considering a strategy targeted at the group with high SEP and middle-low education level. **i** Probability of success of the targeted strategy, defined as the fraction of contact matrices for which there exists a critical control effort allowing epidemic control, out of a random sample of 10,000 synthetic expanded contact matrices with various assortativity levels. **j** Pie charts displaying the distribution of the population in three partitions, i.e., low SEP/high SEP (top), middle-low education/high education (center), and the combination of the two dimensions (bottom).

young adults. This could be due to differences in lifestyle and habits. Area-based socio-economic factors are often linked to mobility patterns[19,60]. Adults living in low-SEP areas may need to commute for reasons related to work, or have less opportunities to work from home[19], potentially engaging more contacts. In seniors, the number of contacts seems to be the same regardless of the SEP area, however, seniors with higher education engage more contacts, potentially signaling the presence of larger social circles. However, these are tentative interpretations, and we acknowledge that other unknown complex relationships could explain the findings in adults and seniors.

To integrate socio-economic factors into epidemic modeling, some studies used contact data to inform matrices stratified only on the participant side[9,61], due to lack of socio-economic information about the contacts. The latter are often not collected through standard contact surveys, and would require detailed contact diaries where participants report additional information about contacts. The absence of these data prevents from uniquely identifying contact assortativity along socio-economic dimensions, and estimating an expanded matrix. Here, we presented a method to partially overcome this issue. Despite relying only on socio-economic

information of the participants, we built a set of expanded contact matrices with varying assortativity levels, by exploiting matrix properties such as reciprocity and aggregation. Notably, we found that the assortativity levels compatible with the data do not span the full parameter space, and are actually rather constrained. This means that socio-economic stratification of participants, combined with information on the age for both participants and contacts, can provide some indication on the full mixing patterns. We found that the large majority of the expanded contact matrices exhibited a higher assortativity level than homogeneous mixing, for both individuals-based and neighborhood-based socio-economic dimensions (education level and SEP). We found that assortativity was more pronounced for SEP than for education level. This could be due to household contacts sharing the same SEP, thus enhancing assortativity in this dimension. Overall, our results are in line with a study on contact data collected in Hungary during the COVID-19 pandemic, which showed preliminary evidence that contacts are assortative along the SES dimension (self-perceived wealth)[34]. Assortativity within socio-economic groups has also been observed in mobility patterns[62].

Accounting for the number of contacts, heterogeneous mixing patterns and population sizes, we found that individuals contributing the most to transmission were children and adults living in high SEP areas. Integrating the matrices into a transmission model revealed heterogeneous disease burden across socio-economic groups, with higher attack rates in adults with high education living in low SEP areas and seniors with high education living in high SEP areas. Notably, these differences were found even under the assumption of equal conditions in susceptibility, infectiousness and disease severity across socio-economic groups. Other health-related heterogeneities across socio-economic groups, for example differences in vaccine uptake, testing behaviour or health barriers, could further exacerbate the differences in epidemic outcomes[20,35,63,64].

The level of contact assortativity is an important factor in determining epidemic dynamics[65]. Using our reconstruction method, we were able to identify ranges of assortativity in education level and SEP dimensions, compatible with the aggregated contact data collected through the survey. We tested the impact of assortativity on the effectiveness of targeted control measures. We found that the higher the assortativity, the lower the chances of a targeted control strategy to be effective, and the higher the control effort needed to contain the epidemic (i.e., a larger proportion of the target group needs to be targeted with immunization or contact reduction measures). Our results are in line with insights from epidemic spreading on networks[66–68]. Assortative networks are still prone to disease spreading despite immunization of part of the network, because they remain highly connected even after removal of some links. Instead, when contacts are less assortative, and closer to homogeneous mixing, effective epidemic control in a subgroup can be beneficial for the whole population, because individuals outside of the target group are less able to sustain the epidemic on their own. One practical public health application of these findings arises in the context of immunization restricted to subgroups of the population, for example, when vaccine hesitancy or unequal access to vaccination reduces coverage in other groups. In such a scenario, a larger allocation of vaccine resources would be required to achieve epidemic control if vaccinated individuals primarily mix within their group, compared to homogeneous mixing.

Several modeling studies have accounted for socio-economic factors indirectly. For example, a COVID-19 modeling study in England used different age-stratified contact matrices for each area based on a spatial index of multiple deprivation, however, matrices were only adapted for population structure, and did not account for different contact activity depending on area-based socio-economic level[61]. Our results suggest that integrating education level and SEP directly into the contact matrix can capture existing heterogeneities in contacts. In this perspective, modeling studies could also benefit from collection of epidemiological data (e.g., incidence and hospitalization) and other behavioral data (e.g., risk perception and mask wearing) stratified by socio-economic status and not only by age, to be integrated into transmission models together with contact data. Moreover, seroprevalence data could be used to fit the assortativity level of the contact matrix[69]. We did not fit our modeling framework to epidemiological data, as such data were not available with the required stratification (i.e., both age and SES or SEP). We acknowledge that stratified data of this kind are rarely accessible, and when available, they often present challenges related to limited sample sizes and privacy concerns.

We acknowledge some limitations of our study. First, participants in the survey did not report the exact neighborhood of residence, but only the municipality. As an approximation, we assigned to each participant the average SEP index of the municipality of residence. This may reduce the granularity of the SEP index among survey participants, however the risk of misclassification is limited, as we only distinguish two groups (high SEP and low SEP based on the median) and also because the SEP index showed limited variation within a municipality. Second, we pulled together data from three distinct survey periods, to reach an adequate sample size while stratifying into 16 groups (age, SEP and education level). We did not perform a longitudinal analysis looking at changes in contact behavior over time, due to small sample sizes. Restrictions during the first survey period

(January – February 2021) were stricter compared to the other survey periods, in terms of stringency index and mobility variations. Third, we used pandemic contact data and pooled participants' responses over several survey waves, therefore, our results may not be generalizable to pre- and post-pandemic behavior. For Switzerland, we did not have pre-pandemic survey data to be used for comparison. A synthetic pre-pandemic age-stratified contact matrix for Switzerland was available[39,70], but it was not stratified by socio-economic factors. Fourth, we considered illustrative control strategies, where only a subgroup of the population was targeted, while the rest of the population was not affected by public health measures. More complex interventions, while potentially more realistic, would have hindered the interpretation of the interplay between assortativity and control effort. Future work could address this aspect, by designing control strategies from an equity perspective. Finally, we generated the expanded contact matrices by distributing the number of contacts across socio-economic groups of the peers, based on an intermediate contact matrix. The latter was estimated from the available data stratifying by age and socio-economic factors on the participants' side. Therefore, any bias in these data could be reflected in the expanded contact matrices. More individual-level data would reduce potential biases and allow to generalize findings to the entire population.

This study uncovers differences in contact behaviour among socio-economic groups so far overlooked, and demonstrates how assortativity in contact patterns, beyond the age dimension, can influence epidemic control. Future characterization of post-pandemic contact behavior accounting for additional socio-economic dimensions can help to inform more effective control measures to prevent infectious disease spread.

## Data availability

The dataset with the SEP index can be requested here[71]. Population and household statistics (STATPOP[72]) 2021-2022 were provided by the Federal Statistical Office under user agreement for the scope of this project. All other population datasets used in this study are openly available from the Federal Statistical Office here[43,44]. Administrative boundaries used in Fig. 1b were retrieved from the Federal Statistical Office[73]. The social contact data, with a subset of variables for participants, are available on Zenodo[40]; the full dataset can be obtained by the authors under request. Requests can be sent to the corresponding author (Laura Di Domenico, laura.didomenico@uhasselt.be). Source data for Figs. 1–5 in the manuscript can be found in Supplementary Data 1, 2, 3, 4 and 5, respectively.

## Code availability

The code used to run the analysis is available on GitHub here[57].

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

## Acknowledgements

This work has received funding from the ESCAPE project (101095619), funded by the European Union, and the Swiss State Secretariat for Education, Research and Innovation (SERI) (22.00482). The work was further supported by the Multidisciplinary Center for Infectious Diseases, University of Bern, Bern, Switzerland. The funders had no role in study design, data collection and analysis, decision to publish, or preparation of the manuscript.

## Author contributions

L.D.D. and C.L.A. conceived and designed the analysis. L.D.D. and M.L.R. cleaned and prepared the data. L.D.D. performed the analysis and wrote the manuscript. All authors critically revised the manuscript and approved its final version.

## Competing interests

The authors declare no competing interests.
