## [Transparent Peer Review file · Communications Medicine]

Individual and neighbourhood based socioeconomic factors relevant for contact behaviour and epidemic control

Corresponding Author: Dr Laura Di Domenico

Version 0:

Reviewer comments:

Reviewer #1

(Remarks to the Author)

Thank you for the opportunity to review this manuscript.

In this work, the authors analyze the relationship between contact patterns relevant to disease transmission and socioeconomic status in Switzerland.

Recently, there has been an increased interest in quantifying the effects of socioeconomic status and social inequalities in disease dynamics, and especially in how these can be addressed through disease modeling.

This study tackles an important and timely research question, with a novel and important contribution that brings new data and new statistical methods to this area.

In particular, the study provides:

- An extensive analysis of contact data from Switzerland, in relation to socioeconomic status, measured in different ways (individual-based and neighborhood-based)
- A new method to build a generalized contact matrix, fully stratified by age and SES. With this method, the authors identify a region in the assortativity space compatible with the data.
- An extensive simulation study of the impact of SES on epidemic dynamics and the effectiveness of targeted interventions.

Overall, I found this paper to be of excellent quality. The research question is clearly articulated, the methodology is appropriate and rigorously applied, and the results are presented in a clear and compelling manner. The discussion effectively situates the findings within the existing literature and offers meaningful insights.

The paper is well-organized, the writing clear and concise, and the overall presentation polished.

The code to reproduce the results is available and well documented.

I did not identify any substantive weaknesses or areas in need of revision. In my view, this manuscript meets the standards for publication in Communications Medicine and makes a valuable contribution to the field that will pave the way to new developments in social epidemiology and disease modeling.

I have only a minor remark that could improve the discussion.

I believe that one major limitation of this study, as is the case for similar ones, is that the modeling framework is not fitted to surveillance data or any other epidemiological outcome. This also represents a challenge for the field, since epidemiological data of infectious diseases are not generally stratified by SES.

This general point is partly addressed on lines 398-402, however, the manuscript could benefit from expanding the paragraph on this.

Reviewer #2

(Remarks to the Author)

Dear Authors,

Thank you for the opportunity to review your manuscript submitted to Communications Medicine. Your study presents an

important and timely contribution by integrating both individual- and neighbourhood-level socio-economic heterogeneity into epidemic modeling. The manuscript is well-organized, methodologically robust, and of high relevance to public health modeling.

After a careful reading, I believe the manuscript would benefit from minor revisions to improve clarity and accessibility for a broad readership. Below, I outline several points for your consideration:

1. Abstract. The abstract can be improved by adding some measures of the impact of integrating SES and SEP into the parameterization of the models.
2. Clarify the SEP Index Definition and Operationalization: While SEP is introduced as a neighbourhood-based measure, its practical implementation using municipality-level averages should be clarified earlier in the main manuscript. Please explicitly and briefly acknowledge how this index is derived and discuss its potential limitations (in the discussion) — particularly with respect to potential misclassification or reduced granularity.
3. Expand Explanation of Assortativity Indices: The concept of contact assortativity is central to your findings, yet its definition and interpretation (e.g., values >1 vs. <1) are not easily accessible to non-specialist readers. I recommend:
 - Including a concise explanation in the Results section.
 - Clarifying the scale and meaning of the indices (0 = disassortative, 1 = homogeneous, 2 = fully assortative).
4. Clarify Distinction Between Attack Rates and Transmission Contribution: Your analysis distinguishes between groups most affected (by attack rate) and those that drive transmission (via contribution to R_0). This is valuable, but the distinction could be better emphasized in the text or figure captions.
5. Construction of expanded contact matrices. I wonder whether the authors could show some kind of validation of the procedure used to construct the expanded matrices. I understand that this depends on the availability of contact data stratified by some socio-economic indicator which may be very difficult to find. I suggest however to look into data provided by Ref. 34 and see whether this could be usable. I think a validation of the methodology could make the results much more robust. At least acknowledge (expand a bit more) the lack of validation as a limitation in the discussion section. Also a bit of comparison between the method used in the paper and the one adopted in Ref.34 to construct synthetic contact matrices adjusted by SES level, may be appropriate to showcase the differences and possible advances here. This manuscript and the one by Manna et al., are the two major ones that are investigating on this very important aspect of disease spread and control.
6. Contextualize Control Strategy Results for Public Health Practice: The section on targeted interventions is well-executed but could be improved in my opinion by adding a clearer interpretation of why less assortativity leads to lower control efforts and how this insight translates to policymaking.
7. Justify Assumptions About Epidemiological Homogeneity: You assume identical susceptibility, infectiousness, and recovery rates across socio-economic groups (except in children in one scenario). This is a strong (and fair I believe) assumption but merits a short justification in the Methods or Discussion. If relevant, please acknowledge how incorporating heterogeneity in health status or vaccine uptake might affect results.
8. Font size in figures captions is too small (fig 3b, 3d)
9. Line 87. I would provide a bit more context and data description before giving the results, given that the Method section is below.
10. Figure 1a. Not very straightforward to read and understand. Not clear that the sum of the bars gives 100%. How about using a waffle chart (<https://github.com/holtzy/The-Python-Graph-Gallery/blob/master/static/graph/web-waffle-chart-as-share.png?raw=true>) with maybe 2 colours to distinguish between SEP levels, and using //// for different educational levels?
11. Figure 1b. Not mentioned in the text. Maybe in line 90 the reference should be to Fig.1b rather than Fig.1a. It would be nice to add the main cities in the map, to understand whether the blue areas are connected to these bigger cities. And then you could add a discussion point on this. Are these areas also the ones with higher prevalence of comorbidities and/or at higher risks of COVID-19? How heterogeneous are the health services and infrastructures across areas? The point on equity and barriers is not very much explored in the discussion. Maybe you could expand a bit on this very important aspect.
12. Lines 217-226. It would be good to add some numbers to showcase the results that are described here. The numbers are shown in the histogram below (Fig 4), but it would be better to have some references in the text as well. For example when it is stated "the relative contribution and the role in transmission of each socio-economic group is highly heterogeneous"  it would be nice to have a range here and, similarly, for the other discussed results.
13. Figure 2e. The reference in the figure to the different survey waves A1, B1, C1, D1, E1 F1 is not clear and I think these labelling is not used in the main text but only in the supmat.
14. Line 126. No bracket after 'middle-low'

Once again, I commend the authors for this thoughtful and methodologically innovative paper. I look forward to reviewing a revised version.

Reviewer #3

(Remarks to the Author)

This manuscript proposes a modelling approach to exploring how heterogeneity in socio-economic status (SES) could impact disease burden and the selection of effective interventions. The study makes use of social contact and socio-economic data from Switzerland (2021), proposing a novel approach to estimating contact matrices stratified by both age and SES.

The manuscript is clearly written and of broad interest: better understanding the implications of social and economic heterogeneity for dynamics and control of infectious diseases, and how these can be incorporated into mathematical models, is highly topical and this study makes a potentially valuable contribution.

I have several comments, suggestions and queries, outlined below.

Comments:

Line 48: While the introduction talks about SES factors in general, the analysis focuses solely on educational level, but this choice is not motivated or justified (over other possible choices such as personal or household income). While the distinguishing feature between “high” and “middle-low” is not defined, this leads to the slightly odd outcome that all children (0-14y) and most young adults (15-24y) are labelled as middle-low education and hence, by association, middle-low SES, which doesn't seem quite right, given that at least some of these children might (a) be on their way to achieving high education, and (b) live in otherwise high-education households, both of which might be presumed to have an impact on their contact patterns. While it is plausible that income and education would be correlated, I do wonder how assigning individual SES at the household level (and so better distinguishing young people based on their immediate environment, vs their broader neighbourhood might affect some of the model results presented later in the manuscript (i.e., Figs 4 & 5). This is my main concern about the design of the study.

Line 93: I found Figure 1 quite difficult to follow. In panel (a) the “pairs” of bars feel a bit arbitrary (although it did highlight the issue noted above!). The y-axis label should indicate the unit (years). In panel (b) I found distinguishing associations between SEP and population very difficult (the pattern of the borders is hard to distinguish from the pattern of the filled areas). Unless the spatial distribution of regions is relevant, this might be better illustrated by a 2D scatter plot of SEP against population. Also, this panel is not currently referred to in the text, though I suspect the reference in line 90 is supposed to be to Fig. 1b.

Line 103: Figure 2, panels (e) and (f) contain a lot of information, only some of which is referred to in the manuscript text. I suggest only presenting the relevant RRs here and perhaps moving the full panels to supplementary material.

Line 138: This would seem an appropriate point at which to discuss your decision to focus on education as the specific SES metric that your analysis focuses on.

Line 228: Figure 4 – what would the epidemic outcomes look like if contact matrices were based solely on age? This seems like an important baseline in order to establish how much difference the contact heterogeneity based on education/SEP introduces. (I wondered if this was equivalent to the pale blue bars in Fig 4 (a) and (b); however, I think these are just aggregated across the various education/SEP groups, rather than computed in the absence of contact heterogeneity.) This query applies equally to the control strategy results shown in Figure 5. As a minor point, using a consistent layout in Figure 1(a) to that used in Figure 4 (ie, independent bars) may be more consistent and hence clearer.

Line 280: This paragraph contains key results but was also the densest and most difficult to read. Please consider splitting into multiple paragraphs such that there is one key observation (plus evidence to support) per paragraph. Also I recommend using explicit descriptions (rather than, eg, “former” which requires the reader to jump around to remember what it was).

Line 302 (Figure 5(i)): Are these the probabilities of success across all possible combinations of assumptions about contact assortativity with respect to education/SEP?

Line 303: “The overall control effort required ranges between 33% and 45%” – this is calculated over the subset of matrices for which control was possible, correct? If so, please clarify in text.

Line 328: “In a sensitivity analysis, this trend was detected...” Is this included in the supplementary material? Please add a reference.

Line 348: “Adults living in low-SEP areas may need to commute for reasons related to work, potentially engaging in more contacts.” Can you cite references to support this hypothesis? It may well be true, but also feels a bit speculative in the absence of support. Our group has looked at (local) data and observed quite complex relationships between SEP (equivalent) and commuting patterns that are difficult to generalise.

Line 372: Are the differences described in this paragraph across the range of assortativity assumptions explored in the study?

Line 403: Can you comment on how variable neighbourhood SEP typically is within any given municipality? (ie, to what extent is this limitation likely to be material).

Line 403: Acknowledging that small numbers limit longitudinal analysis, can you comment on general social-epidemiological factors that may have influenced contact behaviour during the periods that the surveys were conducted? (eg, were mobility/stay-at-home restrictions in place? This would presumably have a strong influence on contact behaviour).

Line 426: The idea of “more ... equitable control measures” is certainly appealing. However, the study hasn’t really been discussed from the perspective of equity. Clearly, the control measures investigated would have substantial equity impacts (!) however, as noted, these are reasonably chosen to be illustrative rather than realistic. If this is to be a final claim of the paper, it would be good to include some discussion of potential implications from an ethical / equity perspective.

Minor comments:

Line 117: I suggest (presuming I have interpreted correctly) “... had more contacts compared to those living in high SEP areas.” Instead of “... had more contacts with respect to those with high SEP.” The descriptions of data/trends in these analyses can be quite dense to parse, so I think it is better to be clearer/consistent in language (if less concise) to aid correct understanding. In general, I think “compared to” would be more accurate than “with respect to” throughout, as you are discussing comparisons between groups rather than connections (which is what “with respect to” implies to me).

Line 456: “neighbourhood-based” -> “neighbourhood-based”

Version 1:

Reviewer comments:

Reviewer #1

(Remarks to the Author)

I thank the authors for addressing my comment in their revision. This is a high-quality study that deserves to be published in Communications Medicine.

Reviewer #2

(Remarks to the Author)

Reviewer #3

(Remarks to the Author)

Thank you for your detailed response to my and other reviewers' comments, and the clarifications, explanations, and additional simulation analyses added to the manuscript, which I feel complement a strong story. I am satisfied that my earlier concerns have been addressed.

Reviewer #1

Thank you for the opportunity to review this manuscript.

In this work, the authors analyze the relationship between contact patterns relevant to disease transmission and socioeconomic status in Switzerland. Recently, there has been an increased interest in quantifying the effects of socioeconomic status and social inequalities in disease dynamics, and especially in how these can be addressed through disease modeling. This study tackles an important and timely research question, with a novel and important contribution that brings new data and new statistical methods to this area. In particular, the study provides:

- An extensive analysis of contact data from Switzerland, in relation to socioeconomic status, measured in different ways (individual-based and neighborhood-based)
- A new method to build a generalized contact matrix, fully stratified by age and SES. With this method, the authors identify a region in the assortativity space compatible with the data.
- An extensive simulation study of the impact of SES on epidemic dynamics and the effectiveness of targeted interventions.

Overall, I found this paper to be of excellent quality. The research question is clearly articulated, the methodology is appropriate and rigorously applied, and the results are presented in a clear and compelling manner. The discussion effectively situates the findings within the existing literature and offers meaningful insights. The paper is well-organized, the writing clear and concise, and the overall presentation polished. The code to reproduce the results is available and well documented. I did not identify any substantive weaknesses or areas in need of revision. In my view, this manuscript meets the standards for publication in Communications Medicine and makes a valuable contribution to the field that will pave the way to new developments in social epidemiology and disease modeling.

I have only a minor remark that could improve the discussion.

1. I believe that one major limitation of this study, as is the case for similar ones, is that the modeling framework is not fitted to surveillance data or any other epidemiological outcome. This also represents a challenge for the field, since epidemiological data of infectious diseases are not generally stratified by SES. This general point is partly addressed on lines 398-402, however, the manuscript could benefit from expanding the paragraph on this.

We thank the reviewer for the very positive assessment of our manuscript. Regarding the limitation mentioned above, we did not fit our modeling framework to epidemiological data, as such data were not available with the required stratification (i.e., both age and SES or SEP). We acknowledge that stratified data of this kind are rarely accessible, and when available, they often present challenges related to limited sample sizes and privacy concerns. We have expanded the corresponding paragraph in the Discussion to address these aspects (lines 450-454): *“We did not fit our modeling framework to epidemiological data, as such data were not available with the required stratification (i.e., both age and SES or SEP). We acknowledge that stratified data of this kind are rarely accessible, and when available, they often present challenges related to limited sample sizes and privacy concerns.”*

Reviewer #2

Dear Authors,

Thank you for the opportunity to review your manuscript submitted to Communications Medicine. Your study presents an important and timely contribution by integrating both individual- and neighbourhood-level socio-economic heterogeneity into epidemic modeling. The manuscript is well-organized, methodologically robust, and of high relevance to public health modeling.

After a careful reading, I believe the manuscript would benefit from minor revisions to improve clarity and accessibility for a broad readership. Below, I outline several points for your consideration.

We thank this reviewer for acknowledging the scientific relevance of our work. We have implemented the suggested minor revisions, which allowed us to improve the clarity of our manuscript.

1. Abstract. The abstract can be improved by adding some measures of the impact of integrating SES and SEP into the parameterization of the models.

To clarify the impact of integrating SES and SEP into model parameterization, we included the following sentence in the abstract: *“Including socio-economic factors into model parameterization had limited effect on the basic reproduction number but substantially influenced the effectiveness of control strategies.”* Additionally, we revised the sentence on the interplay between assortativity and control effort to strengthen the public health implications of our work.

2. Clarify the SEP Index Definition and Operationalization: While SEP is introduced as a neighbourhood-based measure, its practical implementation using municipality-level averages should be clarified earlier in the main manuscript. Please explicitly and briefly acknowledge how this index is derived and discuss its potential limitations (in the discussion) — particularly with respect to potential misclassification or reduced granularity.

We have now clarified earlier in the manuscript (Results section, lines 92-94) that, in absence of information on the residential address, the SEP level for survey participants was assigned based on the municipality of residence, as originally detailed in the Methods (lines 519-525).

This limitation was briefly mentioned in our original submission. We have now expanded the related paragraph in the Discussion, including the following sentence (lines 455-460): *“Participants in the survey did not report the exact neighbourhood of residence, but only the municipality. As an approximation, we assigned to each participant the average SEP index of the municipality of residence. This may reduce the granularity of the SEP index among survey participants, however the risk of misclassification is limited, as we only distinguish two groups (high SEP and low SEP based on the median) and also because the SEP index showed limited variation within a municipality.”*

3. Expand Explanation of Assortativity Indices: The concept of contact assortativity is central to your findings, yet its definition and interpretation (e.g., values >1 vs. <1) are not easily accessible to non-specialist readers. I recommend:

- Including a concise explanation in the Results section.
- Clarifying the scale and meaning of the indices (0 = disassortative, 1 = homogeneous, 2 = fully assortative).

We thank the reviewer for the suggestion. The assortativity index was mathematically defined in the Methods (lines 600-615). We have now included a reader-friendly explanation in the Results section (lines 183-190). *“The assortativity index ranges between 0 and 2, with a value of 1 indicating homogeneous mixing, where contacts between socio-economic groups occur at random in proportion to group sizes. Values above or below 1 indicate assortative mixing or disassortative mixing, respectively, depending on whether contacts occur more frequently within the same socio-economic group or between groups with different socio-economic levels. Values equal to 2 or 0 indicate fully assortative mixing (i.e., only within-group contacts, corresponding to a diagonal matrix) or fully disassortative mixing (i.e., no within-group contacts, corresponding to an empty diagonal), respectively.”* We also added a text label in Fig. 3f indicating homogeneous mixing. These revisions will help non-specialist readers interpret the assortativity index more easily.

4. Clarify Distinction Between Attack Rates and Transmission Contribution: Your analysis distinguishes between groups most affected (by attack rate) and those that drive transmission (via contribution to R_0). This is valuable, but the distinction could be better emphasized in the text or figure captions.

We clarified the distinction earlier in the Results (lines 232-234): *“We characterized disease spreading by identifying which groups generate most infections, and which groups experience the highest infection burden.”*

5. Construction of expanded contact matrices. I wonder whether the authors could show some kind of validation of the procedure used to construct the expanded matrices. I understand that this depends on the availability of contact data stratified by some socio-economic indicator which may be very difficult to find. I suggest however to look into data provided by Ref. 34 and see whether this could be usable. I think a validation of the methodology could make the results much more robust. At least acknowledge (expand a bit more) the lack of validation as a limitation in the discussion section. Also a bit of comparison between the method used in the paper and the one adopted in Ref.34 to construct synthetic contact matrices adjusted by SES level, may be appropriate to showcase the differences and possible advances here. This manuscript and the one by Manna et al., are the two major ones that are investigating this very important aspect of disease spread and control.

We thank the reviewer for the suggestion about the validation. We used the data provided by Ref. 34 to carry out a validation analysis; we included the results in a section in the Supplementary Information (Section 5: Validation). The results of the validation are discussed in the revised manuscript (lines 598-599 in the Methods, lines 199-202 in the Results): *“As a validation, we applied our method on a dataset of social contacts stratified by age and one socio-economic dimension, for which assortativity was known. We showed that*

our method successfully reconstructs a range of assortativity that includes the true value. Results are shown in Fig. S17, S18 and S19 in the Supplementary Information.”

We summarise the results here below.

In their Zenodo repository (<https://zenodo.org/records/12533501>), the authors of Ref. 34 provide the following empirical contact matrix, stratified by 8 age groups and by 3 SES levels.

Analogously to our main analysis, we aimed to consider 4 age groups and 2 SES levels. Hence, we aggregated the provided matrix in 4 age groups (0-14, 15-29, 30-69, 70+), and in 2 SES levels, low SEP (group 1) and high SES (merging group 2 and 3). We acknowledge that our methodology provides flexibility to be extended to a larger number of age groups and SES groups, but we leave this to future work. Based on this stratification, we then computed the age-stratified matrix, the SES-stratified matrix, the fully stratified matrix, and the intermediate matrix where SES information is missing for contacts. Results are displayed below.

The matrix in panel a represents the “true” expanded matrix, which however is not available when SES information on contacts is missing. In our revised manuscript, we now show that we can reconstruct the true expanded matrix from the intermediate contact matrix (panel b). By applying our methodology, we built a candidate set of expanded contact matrices, proving that the true expanded contact matrix is among them.

Given that we only have one additional socio-economic dimension (while in our original analysis we had both education level and SEP), the algorithm for the synthetic expansion

requires a lower number of free parameters compared to the one presented in our main study.

For each of the four diagonal blocks (one for each age group), composed of 4 unknowns, we can write down 3 conditions, i.e. two conditions for the aggregation and one condition on the reciprocity. We then defined one assortativity parameter in the SES dimension, called $q_{v_1}^i \in (0, 1)$, $v_1 = low\ SES$, representing the proportion of contacts that one individual with low SES level and age group i engages with the same group.

For each of the six coupled off-diagonal blocks with 8 variables, corresponding to a couple of age groups i and j with $i \neq j$, we can write 3 conditions on reciprocity, and 4 conditions on the aggregation. We then defined as free parameter $q_{v_1}^{ij} \in (0, 1)$, $v_1 = low\ SES$, similarly to what was done before, representing the proportion of contacts that one individual with low SES level and age group i engages with the group with low SES level and age group j .

In total, we have 10 free parameters. We explored values ranging between 0 and 1 through random sampling from a uniform distribution, for each of the free parameters. We selected only those combinations leading to a positive contact matrix. The resulting selected distributions are illustrated below. For each parameter, we show the median value, the value expected from proportional mixing, and the values computed from the “true” empirical contact matrix. We verified that the latter values, when fed into our algorithm, yield to the true empirical matrix, as expected.

Finally, we generated 10,000 synthetic expanded contact matrices, by sampling from the selected combinations of free parameters. We then computed the ratio in the dominant eigenvalue compared to an age-only stratified matrix, and the global assortativity index. We found that, as expected, our procedure allowed us to reconstruct ranges of assortativity values (panel b, below) compatible with the “true” matrix (which had SES assortativity equal to 1.6).

6. Contextualize Control Strategy Results for Public Health Practice: The section on targeted interventions is well-executed but could be improved in my opinion by adding a clearer interpretation of why less assortativity leads to lower control efforts and how this insight translates to policymaking.

We expanded the section in the Discussion including some insights for policy making (lines 435-440), by adding *“One practical public health application of these findings arises in the context of immunization restricted to subgroups of the population, for example when vaccine hesitancy or unequal access to vaccination reduces coverage in other groups. In such a scenario, a larger allocation of vaccine resources would be required to achieve epidemic control if vaccinated individuals primarily mix within their group, compared to homogeneous mixing.”*

7. Justify Assumptions About Epidemiological Homogeneity: You assume identical susceptibility, infectiousness, and recovery rates across socio-economic groups (except in children in one scenario). This is a strong (and fair I believe) assumption but merits a short justification in the Methods or Discussion. If relevant, please acknowledge how incorporating heterogeneity in health status or vaccine uptake might affect results.

We assumed identical epidemiological characteristics across socio-economic groups, to be able to disentangle the impact of heterogeneity in contacts and heterogeneity in epidemiological parameters. Assuming the same parameters for all socio-economic groups allowed us to derive and interpret the differences in epidemic outcomes as solely driven by contact patterns, which was the focus of our study. This justification is now mentioned in the Methods (lines 645-646). We expect that incorporating heterogeneity in vaccine uptake could further exacerbate the differences in epidemic outcomes, as found in Manna et al., 2024 (Ref. 35 in the manuscript). This aspect has been addressed in the Discussion (lines 421-423).

8. Font size in figures captions is too small (fig 3b, 3d).

Thank you for the comment. We increased the font size of the tick labels in Fig. 3 where possible.

9. Line 87. I would provide a bit more context and data description before giving the results, given that the Method section is below.

We thank the reviewer for the suggestions. We clearly outlined the steps of our analysis in the previous paragraph (lines 71-86). We believe this provides enough context to introduce the Results section.

10. Figure 1a. Not very straightforward to read and understand. Not clear that the sum of the bars gives 100%. How about using a waffle chart (<https://github.com/holtzy/The-Python-Graph-Gallery/blob/master/static/graph/web-waffle-chart-as-share.png?raw=true>) with maybe 2 colours to distinguish between SEP levels, and using //// for different educational levels?

We thank the reviewer for the comment. Also Reviewer #3 found Fig. 1a hard to understand. For this reason, we modified the visualization, adopting a simple bar chart, analogously to what was done in Fig. 4 for the epidemic quantities.

11. Figure 1b. Not mentioned in the text. Maybe in line 90 the reference should be to Fig.1b rather than Fig.1a. It would be nice to add the main cities in the map, to understand whether the blue areas are connected to these bigger cities. And then you could add a discussion point on this. Are these areas also the ones with higher prevalence of comorbidities and/or at higher risks of COVID-19? How heterogeneous are the health services and infrastructures across areas? The point on equity and barriers is not very much explored in the discussion. Maybe you could expand a bit on this very important aspect.

We thank the reviewer for spotting the missing reference to Fig.1b in line 90. As already mentioned in the introduction (lines 53-60), previous work uncovered the association of the SEP index with health-related outcomes such as rate of testing, infection, hospitalization, and mortality related to COVID-19 (Riou et al., Lancet Public Health 2021). We now mention health-related heterogeneities in the Discussion (lines 421-423).

12. Lines 217-226. It would be good to add some numbers to showcase the results that are described here. The numbers are shown in the histogram below (Fig 4), but it would be better to have some references in the text as well. For example when it is stated "the relative contribution and the role in transmission of each socio-economic group is highly

heterogeneous"  it would be nice to have a range here and, similarly, for the other discussed results.

We thank the reviewer for the suggestion. In the revised version of the manuscript, we now provide some numbers in the text (lines 240-254), in relation to the quantities displayed in Fig. 4.

13. Figure 2e. The reference in the figure to the different survey waves A1, B1, C1, D1, E1 F1 is not clear and I think these labelling is not used in the main text but only in the supmat.

We added a sentence in the figure caption explaining the notation.

14. Line 126. No bracket after 'middle-low'

Corrected.

Once again, I commend the authors for this thoughtful and methodologically innovative paper. I look forward to reviewing a revised version.

We greatly appreciate the reviewer's positive feedback on our work. In the revised version, we incorporated their suggestions by including additional results on the validation of our methodology, expanding several paragraphs in the Results and Discussion, and improving the clarity of the figures.

Reviewer #3

This manuscript proposes a modelling approach to exploring how heterogeneity in socio-economic status (SES) could impact disease burden and the selection of effective interventions. The study makes use of social contact and socio-economic data from Switzerland (2021), proposing a novel approach to estimating contact matrices stratified by both age and SES.

The manuscript is clearly written and of broad interest: better understanding the implications of social and economic heterogeneity for dynamics and control of infectious diseases, and how these can be incorporated into mathematical models, is highly topical and this study makes a potentially valuable contribution.

I have several comments, suggestions and queries, outlined below.

We thank the reviewer for their positive assessment of our manuscript. We provide a point-by-point reply to their comments below.

Comments

1. Line 48: While the introduction talks about SES factors in general, the analysis focuses solely on educational level, but this choice is not motivated or justified (over other possible choices such as personal or household income). While the distinguishing feature between “high” and “middle-low” is not defined, this leads to the slightly odd outcome that all children (0-14y) and most young adults (15-24y) are labelled as middle-low education and hence, by association, middle-low SES, which doesn’t seem quite right, given that at least some of these children might (a) be on their way to achieving high education, and (b) live in otherwise high-education households, both of which might be presumed to have an impact on their contact patterns. While it is plausible that income and education would be correlated, I do wonder how assigning individual SES at the household level (and so better distinguishing young people based on their immediate environment, vs their broader neighbourhood might affect some of the model results presented later in the manuscript (i.e., Figs 4 & 5). This is my main concern about the design of the study.

The first part of our study (regression analysis) considered both education level and household income. For the second part (construction of the contact matrix), we focused solely on education level due to the population data available. Indeed, in order to apply our methodology, we need information on the number of individuals by SES level, in each age group and SEP group, in the general Swiss population. We were able to find this information only for SES being education level, and not for household income. We agree with the reviewer that this choice was not justified in the original manuscript. We now mention this in the revised manuscript (Methods, lines 563-566): “As SES indicator, we focused on education level only rather than household income, as stratified population data (i.e., number of individuals \$N_s\$ for any group \$s=(i,v,d)\$ in the general Swiss population) were only available for \$d\$ being education level and not for household income.”

Similarly, the choice of classifying all children and most young adults as middle-low education was again driven by available population data. The distinguishing feature between high and middle-low was defined in lines 498-501. To assign education level to young people based on the education level of their parents, we would need to know which fraction of young individuals in Switzerland (further stratified by SEP) have parents with high or middle-low education. This information was not available in the population statistics. We also note that this distribution may differ from the one observed in the general population of adults, which may or may not have children, and the number of children may also depend on the education level. Nevertheless, as an approximation, we can assume that this distribution is the same as the distribution of high / middle low education in the general adult population. Under this assumption, we carried out a sensitivity analysis where the education level for young individuals was assigned based on their parents, to reflect their immediate environment. We report the results below. The results have been included in an additional section of the Supplementary Information (Section 6: Alternative definition of education level for children).

Regression analysis

For this analysis, we did not have to make any assumption, because we can directly use the information of the participants in the survey. Indeed, parents reporting contacts for their children declare their education level in the survey. We thus assigned to participants below 18 years of age the education level declared by their parents, and we ran the regression analysis again.

We found that the RR of contacts in children with high education compared to low education was not significant. This finding suggests that contacts in children are not highly influenced by the education level of their parents.

Construction of the expanded contact matrix

For this analysis, we distributed the number of low-SEP children (0-14 y.o.) and low-SEP young adults (15-24 y.o., with middle-low education) into middle-low and high education, based on the fraction of low SEP adults with middle-low or high education. Similarly, we distributed the number of high-SEP children and high-SEP young adults (with middle-low

education) into middle-low and high education, based on the fraction of high-SEP adults with middle-low or high education. As a result, the population sizes were updated as follows (to be compared with Fig. 1a for the main analysis).

We then estimated the intermediate contact matrix, and generated a set of synthetic expanded contact matrices compatible with the observed matrix. We illustrate one of the expanded synthetic matrices below. Analogously to the main analysis, we computed the assortativity index and the ratio of the dominant eigenvalue compared to an age-only stratified matrix. We found that the assortativity plane compatible with the data had a similar shape compared to the main analysis, with a slightly bigger area spanning in the bottom right quadrant (lower values in assortativity index). This makes sense as distributing children in two groups - rather than forcing them to be in the same education level - allows to derive compatible matrices with less assortative contacts. The variation observed in R_0 was slightly larger than the main analysis, but still quite small (around 2% rather than 0.7%).

Epidemic spread in absence of control strategies

We provide the results of the sensitivity analysis below (to be compared with Fig. 4). We found that the relative attack rates did not substantially change compared to the main analysis. In terms of contributions to R_0 , children and young adults in middle-low education contribute less in the sensitivity analysis, compared to the main analysis, as expected as their population size is reduced. For the other age groups, results were robust compared to the main analysis.

Effectiveness of targeted control strategies

We provide the results of the sensitivity analysis below (to be compared with Fig. 5), in the scenario with reduced susceptibility in children. As expected, the higher the assortativity in the contacts, the higher the control effort of a targeted strategy required to control the epidemic. A notable difference in the sensitivity analysis compared to the main analysis is the effectiveness of targeting individuals in the group of high SEP and high education. Indeed, we found a set of contact matrices (however very few, less than 5%), for which a control effort around 30% would be effective. In contrast, in the main analysis we found that control was never achievable regardless of the assortativity in contacts. However, this is expected given that the size of the target group (high SEP and high education) is larger in the sensitivity analysis (32%) because it includes also a fraction of the children group, compared to the main analysis (24%) where children were excluded as they were all classified as middle-low education.

2. Line 93: I found Figure 1 quite difficult to follow.

- In panel (a) the “pairs” of bars feel a bit arbitrary (although it did highlight the issue noted above!). The y-axis label should indicate the unit (years).

Reviewer #2 also found Fig. 1a difficult to follow. We revised the visualization using independent bars, as in Fig. 4.

- In panel (b) I found distinguishing associations between SEP and population very difficult (the pattern of the borders is hard to distinguish from the pattern of the filled areas). Unless the spatial distribution of regions is relevant, this might be better illustrated by a 2D scatter plot of SEP against population. Also, this panel is not currently referred to in the text, though I suspect the reference in line 90 is supposed to be to Fig. 1b.

We revised the figure, using a grey gradient for the population sizes rather than a blue/red gradient. The blue and red borders are now much easier to distinguish. We also corrected the reference to the figure in line 90, as spotted by Reviewer #2.

3. Line 103: Figure 2, panels (e) and (f) contain a lot of information, only some of which is referred to in the manuscript text. I suggest only presenting the relevant RRs here and perhaps moving the full panels to supplementary material.

We chose the keep panel (e) in the main text, to show the full set of variables used to derive the adjusted RRs. We agree with the reviewer that a lot of this information was not referred to in the text. In our revised manuscript, we now mention these results, in particular the association of contacts with household size, day of the week and employment status (lines 142-146): *“Finally, we found that contacts significantly increased with household size, as expected (Fig. 2e). Fewer contacts were estimated during weekends compared to weekdays (RR = 0.84 (0.82-0.87)), and for unemployed individuals compared to employed (RR = 0.78 (0.75-0.81)), while the number of contacts did not vary significantly with gender, region, country of birth and vaccination status (Fig. 2e).”*

4. Line 138: This would seem an appropriate point at which to discuss your decision to focus on education as the specific SES metric that your analysis focuses on.

The decision to focus on education level has been justified in the reply to Comment #1 of this reviewer. We have included this explanation in lines 148-150 in the revised manuscript: *“In the rest of the analysis, we focused on education level as SES indicator, as population data were not available for other SES indicators (see Methods).”*

5. Line 228: Figure 4 – what would the epidemic outcomes look like if contact matrices were based solely on age? This seems like an important baseline in order to establish how much difference the contact heterogeneity based on education/SEP introduces. (I wondered if this was equivalent to the pale blue bars in Fig 4 (a) and (b); however, I think these are just aggregated across the various education/SEP groups, rather than computed in the absence

of contact heterogeneity.) This query applies equally to the control strategy results shown in Figure 5.

We thank the reviewer for the comment. We confirm that the pale blue bars in Fig. 4 (a) and (b) refer to attack rates aggregated across the various education/SEP groups.

We have now computed all epidemic outcomes in Fig. 4 and Fig. 5 assuming an age-only stratified matrix, or equivalently, a matrix stratified by age, SEP and education level but assuming homogeneous mixing in education and SEP dimensions. Results are summarised below.

Epidemic spread in absence of control strategies

In Fig. 4, we added horizontal dashed lines indicating the relative attack rate and the contribution to R_0 with a model assuming homogeneous mixing in the SEP and education level. We updated the figure caption accordingly. Concerning attack rates (Fig. 4a,c), we found that incorporating heterogeneity in contacts had limited impact on attack rates overall and by age group compared to a homogeneous model, (as expected given the small differences in the largest eigenvalues of the matrices, Fig. 3e). However, incorporating heterogeneous mixing patterns allowed to uncover differences in attack rates across socio-economic groups (which would otherwise be homogeneous if using a contact matrix based solely on age). Concerning the contribution to R_0 (Fig. 4b,d), considerable differences were visible compared to the reference model, for example young adults with low SEP contributed more compared to homogeneous mixing, compensating for a reduced contribution in those with high SEP.

Effectiveness of targeted control strategies

We computed the type-reproduction number and the corresponding immunity threshold for the set of target groups of the main analysis, using an expanded contact matrix assuming homogeneous mixing in the education level and SEP dimensions. The immunity thresholds

are now reported in Fig. 5 and Fig. S16 as vertical dashed lines. As expected, the control effort required in the homogeneous mixing scenario is lower, compared to scenarios with higher contact assortativity. We updated Fig. 5 and the text in the caption accordingly, and Fig. S16 in the Supplementary Information.

As a minor point, using a consistent layout in Figure 1(a) to that used in Figure 4 (ie, independent bars) may be more consistent and hence clearer.

We followed the reviewer’s suggestion and adapted Fig. 1a using independent bars (see Comment #2).

6. Line 280: This paragraph contains key results but was also the densest and most difficult to read. Please consider splitting into multiple paragraphs such that there is one key observation (plus evidence to support) per paragraph. Also I recommend using explicit descriptions (rather than, eg, "former" which requires the reader to jump around to remember what it was).

We thank the reviewer for their helpful suggestions. We largely revised the paragraph and splitted the information into multiple subparagraphs, to improve the clarity of the presentation of the results.

7. Line 302 (Figure 5(i)): Are these the probabilities of success across all possible combinations of assumptions about contact assortativity with respect to education/SEP?

Yes, to be more precise, we generated 10,000 realizations of the synthetic expanded contact matrices, by sampling at random the values of the free assortativity parameters from the distributions identified to be compatible with the data. We have now clarified this in the figure caption, by writing “Probability of success of the targeted strategy, defined as the fraction of contact matrices for which there exists a critical control effort allowing epidemic control, out

of a random sample of 10,000 synthetic expanded contact matrices with various assortativity levels.” (lines 308-310).

8. Line 303: “The overall control effort required ranges between 33% and 45%” – this is calculated over the subset of matrices for which control was possible, correct? If so, please clarify in text.

Yes, this is correct. We clarified this in the text in lines 295-297 and lines 343-344.

9. Line 328: “In a sensitivity analysis, this trend was detected...” Is this included in the supplementary material? Please add a reference.

Done.

10. Line 348: “Adults living in low-SEP areas may need to commute for reasons related to work, potentially engaging in more contacts.” Can you cite references to support this hypothesis? It may well be true, but also feels a bit speculative in the absence of support. Our group has looked at (local) data and observed quite complex relationships between SEP (equivalent) and commuting patterns that are difficult to generalise.

We agree with the reviewer that this is a tentative interpretation, as other complex underlying relationships could explain the observed result. We revised the text, highlighting the challenges in interpreting this finding (lines 393-394). On this note, we cited again Ref. 19 (Jay et al., Nat. Hum. Behav. 2020), where authors found that during the COVID-19 pandemic “*residents of low-income neighbourhoods were more likely to work outside the home, compared to residents in higher-income neighbourhoods, but were not more likely to visit locations such as supermarkets, parks and hospitals.*”

11. Line 372: Are the differences described in this paragraph across the range of assortativity assumptions explored in the study?

Yes, this paragraph refers to epidemic outcomes obtained using a set of expanded contact matrices with various assortativity assumptions (Fig. 4).

12. Line 403: Can you comment on how variable neighbourhood SEP typically is within any given municipality? (ie, to what extent is this limitation likely to be material).

For each municipality, we computed the coefficient of variation ($100 \times \text{standard deviation} / \text{mean}$) as a measure of dispersion of the neighbourhood SEP, and we compared it with the coefficient of variation of the full SEP distribution (16%). We found that more than 96% of municipalities had a lower coefficient of variation, with 57% having CV lower than 10%, indicating that neighborhood SEP showed limited variation within municipalities. We commented on this in lines 458-460.

13. Line 403: Acknowledging that small numbers limit longitudinal analysis, can you comment on general social-epidemiological factors that may have influenced contact behaviour during the periods that the surveys were conducted? (eg, were

mobility/stay-at-home restrictions in place? This would presumably have a strong influence on contact behaviour).

To provide context on the survey periods, we looked at the stringency index and mobility variations in Fig. S1. The period of January - February 2021 was characterized by a higher stringency index and a stronger variation in mobility, meaning that public health measures were stricter compared to the other survey periods. We now mention this in the Discussion (lines 463-465).

14. Line 426: The idea of “more ... equitable control measures” is certainly appealing. However, the study hasn’t really been discussed from the perspective of equity. Clearly, the control measures investigated would have substantial equity impacts (!) however, as noted, these are reasonably chosen to be illustrative rather than realistic. If this is to be a final claim of the paper, it would be good to include some discussion of potential implications from an ethical / equity perspective.

We agree that our manuscript does not fully address the equity perspective. We revised the last paragraph, leaving the design of equitable control measures for future work (lines 472-473).

Minor comments

15. Line 117: I suggest (presuming I have interpreted correctly) “... had more contacts compared to those living in high SEP areas.” Instead of “... had more contacts with respect to those with high SEP.” The descriptions of data/trends in these analyses can be quite dense to parse, so I think it is better to be clearer/consistent in language (if less concise) to aid correct understanding. In general, I think “compared to” would be more accurate than “with respect to” throughout, as you are discussing comparisons between groups rather than connections (which is what “with respect to” implies to me).

We thank the reviewer for the suggestion. We agree that using “compared to” is more accurate than “with respect to”. We replaced the expression throughout the manuscript.

16. Line 456: “neighbourhoooh-based” -> “neighbourhood-based”

Typo corrected.